# Extreme value statistics of edge currents in Markov jump processes and their use for entropy production estimation

**Izaak Neri[1]⋆ and Matteo Polettini[2]†**

**1** Department of Mathematics, King's College London, Strand, London, WC2R 2LS, UK
**2** Department of Physics and Materials Science, University of Luxembourg,
Campus Limpertsberg, 162a avenue de la Faïencerie,
L-1511 Luxembourg (G. D. Luxembourg)

⋆ izaak.neri@kcl.ac.uk , † matteo.polettini@uni.lu

## Abstract

The infimum of an integrated current is its extreme value against the direction of its average flow. Using martingale theory, we show that the infima of integrated edge currents in time-homogeneous Markov jump processes are geometrically distributed, with a mean value determined by the effective affinity measured by a marginal observer that only sees the integrated edge current. In addition, we show that a marginal observer can estimate a fraction of the average entropy production rate in the underlying nonequilibrium process from the extreme value statistics in the integrated edge current. The estimated average rate of dissipation obtained in this way equals the above mentioned effective affinity times the average edge current. Moreover, we show that estimates of dissipation based on extreme value statistics can be significantly more accurate than those based on thermodynamic uncertainty ratios, as well as those based on a naive estimator obtained by neglecting nonMarkovian correlations in the Kullback-Leibler divergence of the trajectories of the integrated edge current.

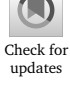

# 1 Introduction

Currents with nonzero average value are a hallmark of nonequilibrium processes. In statistical physics, there has been much interest in characterising the statistics of currents, with initial work focusing on fluctuation relations [2–4]. More recently, it was shown that the large deviation rate function of a current is upper bounded by a parabola with a prefactor that is proportional to the entropy production rate [5,6] and that the Fano factor of currents is bounded from below by the inverse dissipation rate [7–9]. Hitherto, current fluctuations have mainly been considered at fixed times. However, since currents are stochastic processes, it is possible to quantify current fluctuations through other properties of a trajectory, such as the first-passage properties [10–19] or the (closely related) extreme value statistics of currents [20].

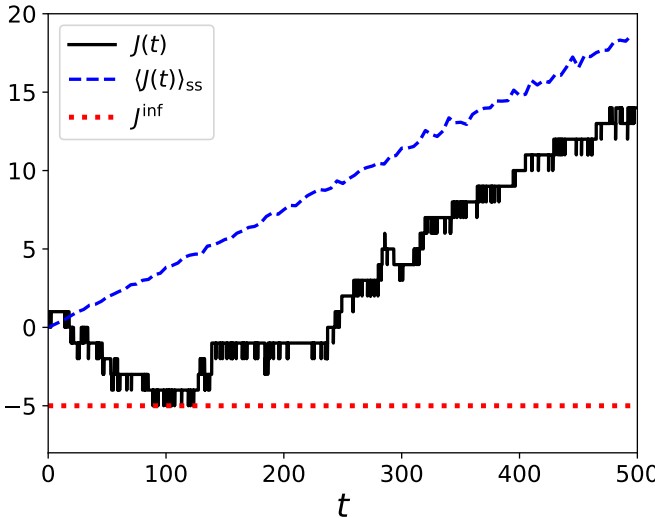

Figure 1: Illustration of the infimum $J^{\mathrm{inf}}$ (dotted, red line) in a trajectory $J(t)$ (black, solid line) of an empirical time-integrated current with positive mean flow $\langle J(t)\rangle_{\mathrm{ss}}$ (blue, dashed line). The data is taken from a trajectory of the position variable $J = J_{\mathrm{pos}}$ in the kinesin-1 model defined in Sec. 7 with parameters $[\mathrm{ATP}] = 0.1\mu\mathrm{M}$ and $f_{\mathrm{mech}} = 3.5\mathrm{pN}$.

In the present paper we focus on this latter. Consider an empirical time-integrated current $J(t)$ in a nonequilibrium stationary state of a classical, stochastic process. The time index $t \in \mathbb{R}^+$, and we use the conventions that the current is zero at the origin of time and increases on average. The infimum $J^{\mathrm{inf}}$ of the current, as illustrated by the red dotted line in Fig. 1, is the most negative value that the current takes, and hence it determines its largest excursion in the direction that opposes the average flow.

Adding to the fact that extreme value statistics of stochastic processes are interesting in their own right, see e.g. Ref. [21], there are also a couple of specific reasons from nonequilibrium thermodynamics why we would like to study extreme values of currents. A first reason is because the statistics of extreme values exhibit universal properties that are analytically tractable with martingale methods [22–25]. For example, for the entropy production $S$, which is one the most well-studied examples of a current, exact results have been derived for the statistics of the infimum value $S^{\mathrm{inf}}$. References [23,24] show that the mean infimum of the entropy production is greater than or equal to one, i.e.,

$$\langle S^{\mathrm{inf}}\rangle_{\mathrm{ss}} \geq -1,\tag{1}$$

and an analogous bound holds for the cumulative distribution of the infimum [23, 24], viz.,

$$\mathbb{P}_{ss}\left(S^{\inf} \leq -\ell\right) \leq e^{-\ell}, \quad \forall \ell \in \mathbb{R}^+, \tag{2}$$

where $\mathbb{P}_{ss}(\cdot)$ is the probability measure in the stationary state and $\langle\cdot\rangle_{ss}$ denotes the average with respect of $\mathbb{P}_{ss}$; when $S(t)$ is continuous the equalities in Eqs. (1)-(2) are attained. Equations (1)-(2) constrain negative fluctuations of the entropy production, which have also been studied in experimental setups [26, 27].

Note that the infimum statistics of entropy production, as given by Eqs. (1)-(2), follow from the fact that $e^{-S}$ is a martingale. This latter property is a direct consequence of the fact that $e^{-S}$ is a density process — also known as the Radon-Nikodym derivative process [23, 28, 29] — relating the statistics of two probability measures, viz., the probability measure $\tilde{\mathbb{P}}_{ss}$ of the time-reversed process and the probability measure $\mathbb{P}_{ss}$ of the forward process. It should be noted that Radon-Nikodym derivative processes of arbitrary measures with respect to $\mathbb{P}_{ss}$ are martingales, see Appendix A. Hence the martingale approach can be extended to processes other than the entropy production, as long as an appropriate probability measure can be identified, see e.g,. Refs. [22, 30, 31].

A second reason why extreme values of currents are interesting is because they can be used to estimate the average rate $\dot{s}$ in stationary processes consisting of variables that have even parity under time reversal, which is the case when inertia is negligible and external forces are not governed by magnetic fields, see Refs. [10, 16, 19]. Indeed, let $J(t)$ be an integrated, empirical current in an overdamped, Langevin process or a Markov jump process, and let $\bar{j} = \langle J(t)\rangle_{ss}/t > 0$ be its rate on average. If we define

$$\hat{s}_{\inf}(\ell) := \frac{|\ln p_-(\ell)|}{\ell}\bar{j}, \tag{3}$$

where

$$p_-(\ell) := \mathbb{P}_{ss}\left(J^{\inf} \leq -\ell\right) \tag{4}$$

is the probability that the infimum value $J^{\inf}$ is smaller or equal than $-\ell$, then

$$\lim_{\ell \to \infty} \hat{s}_{\inf}(\ell) \leq \dot{s}. \tag{5}$$

The equality here is attained when $J$ is proportional to $S$. Note that although the relations Eqs. (3) and (5) did not appear before in the literature, they can be seen as a particular case of the first-passage ratio $\hat{s}_{FPR}$ in Ref. [19] when the positive threshold of the associated first-passage problem diverges. Interestingly, far from equilibrium, $\hat{s}_{\inf}(\ell)$, for large values of $\ell$, captures a finite fraction of the total rate of entropy production, while estimates based on the variance of the current capture a negligible fraction of the total entropy [19]. Hence, far from equilibrium, it is more effective to use extreme values of currents to estimate the rate $\dot{s}$ of entropy production than to use the variance of the current.

In the present paper, we study in detail the infima statistics of empirical, integrated, *edge currents* in stationary, Markov jump processes, i.e., currents along the edges of a graph representing the different possible transitions in state space. Following Refs. [32, 33], we obtain a martingale that is closely related to the edge currents of a Markov jump process and that is the Radon-Nikodym derivative process of a measure, which we call $\mathbb{R}_{ss}$, with respect to $\mathbb{P}_{ss}$. Subsequently we use martingale manipulations, similar to those presented in Ref. [24] for the exponentiated negative entropy production, to determine the statistics of extreme values of edge currents.

In stationary processes with variables that have even parity under time reversal, the derived results for the extreme value statistics of edge currents can be interpreted in terms of an effective thermodynamic picture of a *marginal observer* that only sees the edge current $J$,

ignorant of the existence of other currents in the system. Such an observer assumes that $J$ is proportional to the entropy production and measures an effective affinity $a(t)$ through the relation [32–34]

$$\langle e^{-a(t)J(t)}\rangle_{ss} = 1.\qquad(6)$$

In this paper we show that: (i) the affinity

$$\lim_{t\to\infty} a(t) = a^*\qquad(7)$$

determines the extreme value statistics of the edge current; (ii) using extreme value statistics, a marginal observer estimates a dissipation rate $\bar{j}a^*$, i.e., we identify

$$\lim_{\ell\to\infty}\frac{|\ln p_-(\ell)|}{\ell} = a^*,\qquad(8)$$

connecting the thermodynamics of the edge current (according to Eqs. (3), (5), and (8)) with its kinematics (according to Eq. (6)). These results show, in line with the results in Refs. [32–35], that although the edge current $J$ is non-Markovian, a marginal observer can develop a consistent, effective thermodynamics.

According to Eq. (5), estimating $\dot{s}$ with $\hat{s}_{\mathrm{inf}}(\ell)$ requires measurements of the cumulative probability $p_-(\ell)$ at large thresholds $\ell$. This is undesirable as the probability $p_-(\ell)$ decays exponentially fast as a function of $\ell$, and therefore it is difficult to empirically estimate $p_-$ at large values of $\ell$; we call this the *infinite threshold* problem [10, 16, 19]. In this paper, for the particular case of integrated edge currents, i.e., $J = J_{x\to y}$, we resolve the infinite threshold problem with the estimator

$$\hat{\overset{\circ}{s}}_{\mathrm{inf}}(\ell) := \bar{j}\ln\frac{p_-(\ell)}{p_-(\ell+1)},\qquad(9)$$

for which we show that

$$\hat{\overset{\circ}{s}}_{\mathrm{inf}}(\ell) = \lim_{\ell'\to\infty}\hat{s}_{\mathrm{inf}}(\ell') = \bar{j}a^* \le \dot{s},\qquad(10)$$

for all $\ell \in \mathbb{N}$. Hence, using the estimator $\hat{\overset{\circ}{s}}_{\mathrm{inf}}(\ell)$ for $\ell \neq 0$, a proportion of the average rate of dissipation, $\dot{s}$, can be estimated from the probability $p_-(\ell)$ that the infimum of an edge current is smaller than a *finite* threshold value $\ell$, resolving the infinite threshold problem.

The quantity $\bar{j}a^*$ has also been studied in Ref. [35], where it is called the average *informed partial* entropy production rate. Reference [35] shows that $\bar{j}a^*$ is a better estimate of dissipation than a naive estimator $\hat{s}_{\mathrm{KL}}$ obtained from neglecting nonMarkovian correlations in the Kullback-Leibler divergence of the trajectories of the current, and which is called the average *passive partial* entropy production rate, i.e., $\dot{s} \ge \bar{j}a^* \ge \hat{s}_{\mathrm{KL}}$. However, Reference [35] argues that $\bar{j}a^*$ cannot be measured passively by observing the trajectory of a current, and instead should be determined as the force at which the edge current stalls, which can be measured actively if we can exert a microscopic force on the system. In the present paper, we show that the informed partial entropy production rate, $\bar{j}a^*$, can be measured passively through the modified infimum ratio $\hat{\overset{\circ}{s}}_{\mathrm{inf}}(\ell)$, as $\hat{\overset{\circ}{s}}_{\mathrm{inf}}(\ell) = \bar{j}a^*$ for $\ell \in \mathbb{N}$.

The paper is organised as follows: We summarise in Sec. 2 the main results. Before addressing the general problem, we derive in Sec. 3 the infimum statistics of an edge current that is proportional to the entropy production; this is a special case that is easily solvable and gives an idea of the results we obtain and mathematical methods we use in the general case. In Sec. 4, we introduce the system setup and some of the mathematical groundwork that we use in later sections to derive the main results. In Sec. 5, following Refs. [32, 33], we introduce a set of martingales associated with the empirical, integrated, edge currents of stationary, Markov jump processes, which constitute the main mathematical tool that permits us to obtain the main results. Subsequently, in Sec. 6, we use the concepts from Secs. 4 and 5 to derive the

main result, which is an explicit expression for the probability mass function of the infima of empirical, integrated, edge currents. In Sec. 7, we show the main result at work on a simple model of two-headed molecular motors [36–38]. In Sec. 8, we study the properties of the estimators $\hat{s}_{\text{inf}}$ and $\mathring{s}_{\text{inf}}$ for the average rate $\dot{s}$ of dissipation based on the infimum statistics of empirical, integrated, edge currents. We end the paper with a discussion in Sec. 9. The paper also contains a few appendices with details about some of the derivations and the model defined in Sec. 7.

## 2 Summary of the main results

We first summarise the main results, and then we discuss how these results are related to the companion paper Ref. [1].

### 2.1 Main results

Let $X(t) \in \mathcal{X}$ be a time-homogeneous Markov jump process, and let

$$J_{x \to y}(t) := N_{x \to y}(t) - N_{y \to x}(t), \quad \text{with} \quad x, y \in \mathcal{X}, \tag{11}$$

denote the difference between the number of times $N_{x \to y}(t)$ that $X$ has jumped from $x$ to $y$ in the time interval $[0, t]$ minus the number of times $N_{y \to x}(t)$ that $X$ has jumped from $y$ to $x$ in the same interval of time. Let us assume, without loss of generality, that $\langle J_{x \to y}(t) \rangle_{\text{ss}} > 0$ when $t$ is large enough.

We call $J_{x \to y}(t)$ an empirical, integrated *edge current*, as it is the flow along the edge $x \to y$ of the graph of possible transitions in the phase space $\mathcal{X}$; we call $x$ the *source* node and $y$ the target node of the edge current. Notice that, for convenience, we often speak of edge currents, tout court, and it should be understood that we consider empirical, integrated currents. Edge currents are the elementary currents of a Markov jump process. Indeed, empirical, time-integrated currents $J$ can be expressed as a linear combination

$$J := \sum_{(u,v) \in \mathcal{E}} c_{u,v} J_{u \to v}(t) \tag{12}$$

of the edge currents, where $c_{u,v}$ is the amount of a certain resource that is exchanged or transported to/from the environment when the process jumps from $u$ to $v$, and where $\mathcal{E} \subset \mathcal{X}^2$ is the set of pairs $(u, v)$ with nonzero transition rates; note that we consider reversible processes for which $(u, v) \in \mathcal{E} \Leftrightarrow (v, u) \in \mathcal{E}$.

The fluctuations of the edge current $J_{x \to y}$ against the direction of the average flow can be characterised by the infimum

$$J_{x \to y}^{\text{inf}} := \inf_{t \geq 0} J_{x \to y}(t). \tag{13}$$

Note that $J_{x \to y}^{\text{inf}}$ is a nonpositive integer as $J_{x \to y}(0) = 0$. If $\langle J_{x \to y}(t) \rangle_{\text{ss}} < 0$, then we can consider the infimum of $-J_{x \to y}$, which is the supremum of $J_{x \to y}$.

In this Paper, using martingale methods similar to those used in Refs. [23, 24] to derive the infimum law Eq. (1), we show that the probability mass function of $J_{x \to y}^{\text{inf}}$ is given by

$$p_{J_{x \to y}^{\text{inf}}}(-\ell | X(0) = x_0) = \begin{cases} e^{-\ell a_{x \to y}^*}(1 - p_{\text{esc}}(x_0))(e^{a_{x \to y}^*} - 1), & \text{if} \quad \ell \in \mathbb{N}, \\ p_{\text{esc}}(x_0), & \text{if} \quad \ell = 0, \end{cases} \tag{14}$$

and its mean value by

$$\langle J_{x \to y}^{\text{inf}} | X(0) = x_0 \rangle_{\text{ss}} = -\frac{1 - p_{\text{esc}}(x_0)}{1 - e^{-a_{x \to y}^*}}, \tag{15}$$

where $a^*_{x\to y} > 0$ is an "effective" affinity that was identified before in Refs. [32–35], and $p_{\mathrm{esc}}(x_0)$ is the probability that the infimum equals zero. In Eqs. (14) and (15) we have used probabilities and expectation values conditioned on a general, initial state $X(0) = x_0$.

In general, $p_{\mathrm{esc}}(x_0)$ does not admit a simple expression in terms of $a^*_{x\to y}$. A notable exception is when $X(0) = x$, where $x$ is the source node of the edge $x \to y$, in which case

$$p_{\mathrm{esc}}(x) = 1 - e^{-a^*_{x\to y}}. \tag{16}$$

When $|a^*_{x\to y}| \ll 1$ the current is stalled at an almost zero average rate, i.e., $\langle J_{x\to y}(t)\rangle_{\mathrm{ss}} \approx 0$, and consequently $p_{\mathrm{esc}}(x_0) \approx 0$, such that the geometric distribution Eq. (14) is approximately an exponential distribution. Equilibrium states are examples of stalled states, but it is also possible to have stalled currents far from equilibrium. Notably, a marginal observer that only measures $J_{x\to y}(t)$ cannot distinguish between an equilibrium state and a nonequilibrium stalled state from the measurements of extreme values of $J_{x\to y}(t)$.

Equation (14) implies that the fluctuations of the extreme values of $J_{x\to y}(t)$ are determined by the effective affinity $a^*_{x\to y}$. The effective affinity $a^*_{x\to y}$ can be defined through the integral fluctuation relation [32–34]

$$\lim_{t\to\infty} \langle e^{-a^*_{x\to y} J_{x\to y}(t)}\rangle_{\mathrm{ss}} = 1, \tag{17}$$

and hence admits a kinematic interpretation. Indeed, applying Jensen's inequality we find

$$a^*_{x\to y}\langle J_{x\to y}(t)\rangle_{\mathrm{ss}} \geq 0. \tag{18}$$

In addition, assuming that the rates $k_{x\to y}$ and $k_{y\to x}$, governing the current of interest, are tunable, then the effective affinity is the difference of their log-ratio $\ln k_{x\to y}/k_{y\to x}$ to values where the average stationary current stalls, $\langle J_{x\to y}\rangle_{\mathrm{ss}} = 0$ [32, 33]. In systems where these rates are regulated by large reservoirs of energy, particles, or (even) information, the effective affinity is the difference of the relevant thermodynamic potentials from the value where the system attains the stalling state. For this reason, Ref. [35] calls $\langle J_{x\to y}(t)\rangle a^*_{x\to y}$ the informed partial entropy production.

As a last result, using Eq. (14), we show that $a^*_{x\to y}$ has a thermodynamic meaning. In particular, we show that $a^*_{x\to y}$ determines the average entropy production rate that a marginal observer estimates from the measurement of the trajectories of $J_{x\to y}$. Indeed, substitution of Eq. (14) in the estimator $\hat{s}_{\mathrm{inf}}$ of $\dot{s}$ — defined in Eq. (3) — we obtain that

$$\lim_{\ell\to\infty} \hat{s}_{\mathrm{inf}}(\ell) = \bar{j}_{x\to y} a^*_{x\to y}, \tag{19}$$

where

$$\bar{j}_{x\to y} := \langle J_{x\to y}(t)\rangle_{\mathrm{ss}}/t \tag{20}$$

is the average current. Hence, according to Eq. (19), $a^*_{x\to y}$ is an effective thermodynamic affinity that when multiplied with the average current rate $\bar{j}_{x\to y}$ determines the entropy production rate $\hat{s}_{\mathrm{inf}}$ measured by a marginal observer. Equation (5) together with Eq. (19) implies

$$\bar{j}_{x\to y} a^*_{x\to y} \leq \dot{s}, \tag{21}$$

an inequality that has also been derived directly from the properties of the generator of the underlying Markov process, as shown in Ref. [33].

Equation (19) is an asymptotic result for large thresholds, as it is based on the estimator $\hat{s}_{\mathrm{inf}}(\ell)$ in the limit of large $\ell$. However, from Eq. (14) it follows that the effective affinity can be estimated using

$$a^*_{x\to y} = \ln \frac{p_-(\ell)}{p_-(\ell+1)}, \tag{22}$$

for all $\ell \in \mathbb{N}$, and hence the effective affinity can be obtained from the measurement of the probability that an edge current goes below a certain *finite* threshold, thus resolving for the case of edge currents the infinite threshold problem [10, 16, 19].

## 2.2 Relation to the companion paper Ref. [1]

The present manuscript comes with the companion manuscript Ref. [1] that addresses similar questions. Reference [1] focuses on the probability

$$\mathfrak{f}_- = 1 - p_{\text{esc}}(x_0) = \mathbb{P}_{\text{ss}}\left(J_{x\to y}^{\inf} \leq -1 | X(0)\right),\tag{23}$$

that the infimum of $J_{x\to y}$ is smaller or equal than $-1$, instead of on its probability mass function. However, the main difference between Ref. [1] and the present manuscript is from a methodological point of view. Reference [1] derives the result Eq. (14) by identifying a Markov process in transition space, while the present manuscript identifies a martingale process associated with $J_{x\to y}$, and subsequently uses this martingale to derive Eq. (14). Both approaches have been developed independently, and consequently both manuscripts can be read independently. Taken together, we believe it is interesting to see how the exact solvability of this problem materialises into two different ways.

Comparing both manuscripts, the following dictionary is useful: Ref. [1] uses $\rho(v|u)$ for the transition rates $k_{u\to v}$, $1 \to 2$ for the observed edge $x \to y$, $c$ for integrated currents $J$, $F$ for the effective affinity $a_{x\to y}^*$, the subindex $\infty$ for stationary states instead of the subindex ss, $\mathfrak{p}_{-n}[p_1^{\mathcal{L}}]$ for the probability mass function $p_{J_{x\to y}^{\inf}}(-\ell | X(0) = x_0)$ of the infimum, and $\mathfrak{p}_0$ for the escape probability $p_{\text{esc}}$.

# 3 Prelude: extreme value statistics of edge currents that are proportional to the entropy production

In this Section, as a simplified initial problem, we derive the statistics of infima of edge currents that are proportional to the entropy production. This problem is relevant both from a mathematical and a physical point of view. From a mathematical point of view, the statistics of currents that are proportional to the entropy production can be determined readily from the fact that $e^{-S}$ is a martingale, see Refs. [23,24], and this constitutes an introduction in a simplified setup to the methods we will use in this paper. From a physical point of view, a marginal observer that only observes the edge current $J_{x\to y}$, unaware of the existence of other currents in the system, thinks that the observed current $J_{x\to y}(t)$ is proportional to the entropy production. Hence, it is insightful to compare the main result Eq. (14) with the infimum statistics of the entropy production.

We consider the entropy production $S(t)$ of a nonequilibrium process that takes the form

$$S(t) = c\, J_{x\to y}(t),\tag{24}$$

where $c > 0$ is a constant, proportionality factor, sometimes called "affinity" from pre-modern alchemic theories of the combination of elements, see Ref. [39].

Consider the stopping problem of establishing the first time entropy production exits a certain interval, i.e.,

$$T := \inf\{t \geq 0 : S(t) \notin (-\ell_- c, \ell_+ c)\}, \quad \text{with} \quad \ell_-, \ell_+ \in \mathbb{N}.\tag{25}$$

Since the interval $(-\ell_- c, \ell_+ c)$ is finite, it holds that

$$p_- + p_+ = 1,\tag{26}$$

where $p_-$ is the probability that both $T < \infty$ and $S(T) \leq -\ell_- c$ hold, and $p_+$ is the probability that both $T < \infty$ and $S(T) \geq \ell_+ c$ hold.

In addition, since $e^{-S}$ is a martingale [22–24], the integral fluctuation relation at stopping times [24]

$$\langle e^{-S(T)}\rangle_{\text{ss}} = 1 \tag{27}$$

applies, and therefore

$$p_-\langle e^{-S(T)}|S(T) \leq -\ell_- c\rangle_{\text{ss}} + p_+\langle e^{-S(T)}|S(T) \geq \ell_+ c\rangle_{\text{ss}} = 1. \tag{28}$$

Since $J_{x\to y}$ changes in increments of size $\pm 1$, the entropy production $S = cJ_{x\to y}$ changes in discrete increments of $\pm c$, and since $S(0) = J_{x\to y}(0) = 0$, Eq. (28) yields

$$p_- e^{\ell_- c} + p_+ e^{-\ell_+ c} = 1. \tag{29}$$

Combining the Eqs. (26) and (29), we obtain that

$$p_- = \frac{1 - e^{-\ell_+ c}}{e^{\ell_- c} - e^{-\ell_+ c}}. \tag{30}$$

In the limit of $\ell_+ \to \infty$, Eq. (30) reduces to

$$p_- = e^{-\ell_- c}, \quad \forall \ell_- \in \mathbb{N}. \tag{31}$$

Identifying

$$p_- = \mathbb{P}_{\text{ss}}\left(S^{\text{inf}} \leq -c\ell_-\right), \tag{32}$$

we obtain for the probability mass function of $S^{\text{inf}}$ [40],

$$p_{S^{\text{inf}}}(-c\ell) = \mathbb{P}_{\text{ss}}\left(S^{\text{inf}} = -c\ell\right) = e^{-\ell c}(1 - e^{-c}), \quad \forall \ell \in \mathbb{N} \cup \{0\}. \tag{33}$$

The average infimum is thus

$$\langle S^{\text{inf}}\rangle_{\text{ss}} = -\frac{c}{e^c - 1} \geq -1, \tag{34}$$

consistent with the infimum law Eq. (1).

Comparing Eqs. (14) and (33), we conclude that, ignoring the prefactor $p_{\text{esc}}(x_0)$ and the value of the probability mass function at $\ell = 0$, a marginal observer measures a statistics for $J_{x\to y}^{\text{inf}}$ that is equivalent to the statistics of the entropy production in a system for which $S = a_{x\to y}^* J_{x\to y}$.

In the following Section we define the system setup in which we will derive Eq. (14) in full generality.

## 4  System setup and mathematical groundwork

We introduce the system setup and some of the mathematical tools that we use to derive the main results.

### 4.1  Markov jump processes

Let $(\Omega, \mathcal{F})$ be a measurable space, with $\Omega$ the set of realisations $\omega \in \Omega$ of a physical process, and $\mathcal{F}$ a $\sigma$-algebra of measurable events. Let $X(\omega, t) = X(t)$, with $\omega \in \Omega$ and $t \in \mathbb{R}^+$ a continuous time index, be a stochastic process defined on $(\Omega, \mathcal{F})$ and that takes values in a finite set $\mathcal{X} \ni X(t)$. Notice that the realisations $\omega$ consist of trajectories over the interval $t \in [0, \infty)$, that $X(\omega, t)$ returns the value of the trajectory at time $t$, and that the set of measurable events contains, amongst others, the sets $\{\omega \in \Omega : X(t, \omega) = x\}$ for all $t \geq 0$ and $x \in \mathcal{X}$. We denote trajectories of $X$ over a finite interval $[0, t]$ by $X_0^t$.

To determine the statistics of $X$, we consider a probability measure $\mathbb{P}_{p^*}$ defined on $(\Omega, \mathscr{F})$, where $p^*$ is the probability distribution of the initial configuration $X(0)$. Notice that probability measures assign probabilities $\mathbb{P}_{p^*}(\Phi) \in [0, 1]$ to events $\Phi \in \mathscr{F}$ in the $\sigma$-algebra $\mathscr{F}$. If we observe trajectories $X_0^t$ in a fixed time interval $[0, t]$, then there is no need to consider the full $\sigma$-algebra $\mathscr{F}$ generated by infinitely long trajectories $X_0^\infty$. Instead, it is then sufficient to consider the sub-$\sigma$-algebra $\mathscr{F}_t$ generated by trajectories $X_0^t$ over a finite time window, and we denote the measure $\mathbb{P}_{p^*}$ restricted to $\mathscr{F}_t$ by $\mathbb{P}_{p^*}[X_0^t]$. In other words, $\mathbb{P}_{p^*}[X_0^t]$ is the probability measure defined on $\mathscr{F}_t$ such that $\mathbb{P}_{p^*}[X_0^t](\Phi) = \mathbb{P}_{p^*}(\Phi)$ for all $\Phi \in \mathscr{F}_t$.

We assume that the pair $(X, \mathbb{P}_{p^*})$ forms a Markov jump process with an initial distribution $p^*(u)$ and rates $k_{u \to v} \geq 0$ (with $u, v \in \mathcal{X}$) that are constant in time $t$. A Markov jump process can be represented by a random walker that moves on the graph $G = (\mathcal{X}, \mathcal{E})$, defined by the vertex set $\mathcal{X}$ and the set of edges

$$\mathcal{E} = \left\{ (u, v) \in \mathcal{X}^2 : k_{u \to v} > 0 \right\}. \tag{35}$$

The probability distribution $p_{X(t)}(u) = p(u; t)$ of $X(t)$, denoting the probability that the random walker is located at time $t$ at $X(t) = u$, solves the differential equation [41]

$$\partial_t p(u; t) = \sum_{v \in \mathcal{X}; v \neq u} p(v; t) k_{v \to u} - p(u; t) \sum_{v \in \mathcal{X}; v \neq u} k_{u \to v}, \quad \forall u \in \mathcal{X}, \tag{36}$$

with initial condition $p(u; 0) = p^*(u)$.

We assume that the directed graph $(\mathcal{X}, \mathcal{E})$ of permissible transitions on which $(X, \mathbb{P}_{p^*})$ is defined is strongly connected, so that the stationary probability mass function $p_{ss}(u)$ that solves

$$p_{ss}(u) = \frac{\sum_{v \in \mathcal{X}; v \neq u} p_{ss}(v) k_{v \to u}}{\sum_{v \in \mathcal{X}; v \neq u} k_{u \to v}} \tag{37}$$

is unique (such Markov jump processes are called irreducible in Ref. [41]). If $p^* = p_{ss}$, then we say that the Markov jump process is *stationary*, and we write $\mathbb{P}_{p^*} = \mathbb{P}_{ss}$. At stationarity, the edge currents, as defined in Eq. (20), are given by

$$\bar{j}_{x \to y} = p_{ss}(x) k_{x \to y} - p_{ss}(y) k_{y \to x}, \tag{38}$$

and the stationarity condition $\partial_t p(u; t) = 0$ implies that

$$\sum_{v \in \mathcal{X}; v \neq u} \bar{j}_{v \to u} = 0, \quad \forall u \in \mathcal{X}. \tag{39}$$

When the *microscopic affinities*

$$a_{u \to v} := \ln \frac{p_{ss}(u) k_{u \to v}}{p_{ss}(v) k_{v \to u}} \tag{40}$$

are equal to zero, i.e.,

$$a_{u \to v} = 0, \quad \forall u, v \in \mathcal{E}, \tag{41}$$

then the Markov jump process $(X, \mathbb{P}_{ss})$ obeys *detailed balance*. For Markov jump processes that obey detailed balance, all edge currents are stalled, i.e., $\bar{j}_{u \to v} = 0$ for all $u, v \in \mathcal{X}$, and we say that the stationary state is an equilibrium state, which we denote by $p_{ss}(u) = p_{eq}(u)$.

We can also represent a Markov jump process in terms of its trajectories $X_0^t = \{X(s): s \in [0, t]\}$. For a Markov jump process, the trajectory $X_0^t$ is uniquely determined by the sequence $(X_0, X_1, \ldots X_{N(t)-1})$ of $N(t)$ states that $X(t)$ occupies in the interval $[0, t]$, and the times $T_i$ when $X(t)$ changed its state from $X_{i-1}$ to $X_i$. Indeed, it holds that

$$X(s) = X_i, \quad \forall s \in [T_i, T_{i+1}). \tag{42}$$

We denote averages of random variables over the measure $\mathbb{P}_{p^*}$ by $\langle \cdot \rangle_{\mathbb{P}_{p^*}}$. If $p^* = p_{ss}$, then we also use $\langle \cdot \rangle_{\mathbb{P}_{ss}} = \langle \cdot \rangle_{ss}$.

## 4.2 Radon-Nikodym derivative processes

Let $\mathbb{Q}_{q^*}$ be a second probability measure defined on $(\Omega, \mathscr{F})$ for which it holds that $(X, \mathbb{Q}_{q^*})$ is a Markov jump process. We denote its initial distribution by $q^*$ and the corresponding transition rates by $\ell_{u \to v} \geq 0$.

We assume that $\mathbb{Q}_{q^*}$ is locally, absolutely continuous with respect to $\mathbb{P}_{p^*}$, i.e.,

$$\mathbb{P}_{p^*}[\Phi] = 0 \Rightarrow \mathbb{Q}_{q^*}[\Phi] = 0 \,, \tag{43}$$

for all $\Phi \in \mathscr{F}_t$ and $t \in \mathbb{R}^+$. Locally refers here to the fact that the two measures are absolutely continuous on the sub-$\sigma$ algebras $\mathscr{F}_t$ for all *finite* $t$, but not necessarily on $\mathscr{F}$. For Markov jump processes, local absolute continuity implies that that

$$k_{v \to u} = 0 \Rightarrow \ell_{u \to v} = 0 \,, \tag{44}$$

and

$$p^*(u) = 0 \Rightarrow q^*(u) = 0 \,, \tag{45}$$

for all $u, v \in \mathcal{X}$.

Since $\mathbb{Q}_{q^*}$ is locally, absolutely continuous with respect of $\mathbb{P}_{p^*}$, there exists a process $R(t)$, which is called the Radon-Nikodym derivative process of $\mathbb{Q}_{q^*}$ with respect to $\mathbb{P}_{p^*}$ [28], such that

$$\langle f(X_0^t) \rangle_{\mathbb{Q}_{p^*}} = \langle f(X_0^t) R(t) \rangle_{\mathbb{P}_{p^*}} \,, \tag{46}$$

for measurable functions $f$ defined on the set of trajectories $X_0^t$. For Markov jump processes, the Radon-Nikodym derivative process takes the form

$$R(t) = \frac{\mathrm{d}\mathbb{Q}_{q^*}[X_0^t]}{\mathrm{d}\mathbb{P}_{p^*}[X_0^t]} = \frac{q^*(X(0))}{p^*(X(0))} \exp\left( \int_0^t dt' \left[ r_p(X(t')) - r_q(X(t')) \right] + \sum_{i=1}^{N(t)-1} \ln \frac{\ell_{X_{i-1} \to X_i}}{k_{X_{i-1} \to X_i}} \right), \tag{47}$$

where

$$r_p(u) = \sum_{v \in \mathcal{X}; v \neq u} k_{u \to v} \quad \text{and} \quad r_q(u) = \sum_{v \in \mathcal{X}; v \neq u} \ell_{u \to v} \tag{48}$$

are the escape rates out of the state $u \in \mathcal{X}$ corresponding to the measures $\mathbb{P}_{p^*}$ and $\mathbb{Q}_{q^*}$, respectively.

If the two measures have the same escape rates, i.e.,

$$r_p(u) = r_q(u), \quad \forall u \in \mathcal{X} \,, \tag{49}$$

then we obtain the simpler expression

$$\frac{\mathrm{d}\mathbb{Q}_{p^*}[X_0^t]}{\mathrm{d}\mathbb{P}_{p^*}[X_0^t]} = \frac{q^*(X(0))}{p^*(X(0))} \exp\left( \sum_{(u,v) \in \mathcal{E}} N_{u \to v}(\omega, t) \ln \frac{\ell_{u \to v}}{k_{u \to v}} \right), \tag{50}$$

where $\mathcal{E}$ is the set of pairs $(u, v) \in \mathcal{X}^2$ so that $\ell_{u \to v} > 0$, and $N_{u \to v}(t)$ is the number of times that $X$ has jumped from $u$ to $v$ in the trajectory $X_0^t$, as used before in Eq. (11). In addition, if

$$\frac{\ell_{u \to v}}{k_{u \to v}} = \frac{k_{v \to u}}{\ell_{v \to u}} \,, \tag{51}$$

then

$$\frac{\mathrm{d}\mathbb{Q}_{p^*}[X_0^t]}{\mathrm{d}\mathbb{P}_{p^*}[X_0^t]} = \frac{q^*(X(0))}{p^*(X(0))} \exp\left( \frac{1}{2} \sum_{(u,v) \in \mathcal{E}} J_{u \to v}(\omega, t) \ln \frac{\ell_{u \to v}}{k_{u \to v}} \right), \tag{52}$$

which is an expression that we use later in the derivation of the main results.

### 4.3 Time-reversal in stationary Markov jump processes and entropy production

Time-reversal arguments play an important role in the derivation of the main results, and it is therefore useful to revise some properties of time-reversal in Markov jump processes.

Let $\Theta$ be the time-reversal map that mirrors trajectories with respect to the origin of time, i.e.,

$$X(\Theta(\omega), t) = X(-t). \tag{53}$$

We define the measure

$$\tilde{\mathbb{P}}_{\text{ss}} = \mathbb{P}_{\text{ss}} \circ \Theta, \tag{54}$$

of time-reversed events. The pair $(X, \tilde{\mathbb{P}}_{\text{ss}})$ is also a stationary Markov jump process with rates

$$\tilde{k}_{u \to v} = k_{v \to u} \frac{p_{\text{ss}}(v)}{p_{\text{ss}}(u)}, \tag{55}$$

and stationary probability mass function $\tilde{p}_{\text{ss}}(u) = p_{\text{ss}}(u)$. Indeed, a direct calculation shows that [42]

$$\frac{d(\mathbb{P}_{\text{ss}} \circ \Theta)[X_0^t]}{d\mathbb{P}_{\text{ss}}[X_0^t]} = e^{-S(t)}, \tag{56}$$

where

$$S(t) = \frac{1}{2} \sum_{(u,v) \in \mathcal{E}} a_{u \to v} J_{u \to v}(t) \tag{57}$$

is the entropy production. Notice that the entropy production is a current of the form Eq. (12) with the coefficients $c_{u \to v}$ given by the microscopic affinities $a_{u \to v}$, as defined in Eq. (40).

If the principle of local detailed balance applies [42–44], which is a statistical physics implementation of local equilibrium [45], then

$$\dot{s} := \langle S(t) \rangle_{\text{ss}} / t = \frac{1}{2} \sum_{(u,v) \in \mathcal{E}} a_{u \to v} \bar{j}_{u \to v} \tag{58}$$

is the entropy production of the second law of thermodynamics, denoting the rate of bits, measured in the natural unit of information (nat), produced on average in the environment. In case the environment consists of one or more thermal reservoirs, then $\dot{s}$ is directly related to the heat dissipated to the environment.

### 4.4 Martingales

Radon-Nikodym derivative processes of the form (47) are martingales with respect to the probability measure $\mathbb{P}_{p*}$ [28].

We use $\mathbb{P}$ to denote a generic probability measure, and not necessarily the measure $\mathbb{P}_{p*}$ of a Markov jump process. A process $M(t)$ is a $\mathbb{P}$-martingale when the following conditions hold: (i) $M(t) = M(X_0^t)$ is a functional on the trajectories of $X$; (ii) $\langle |M(t)| \rangle_{\mathbb{P}} < \infty$; and (iii) the process is driftless, i.e.,

$$\langle M(t) | X_0^s \rangle_{\mathbb{P}} = M(s), \tag{59}$$

for all $s \in [0, t]$.

Martingales are useful for studying properties of processes at random times. This is due to Doob's optional stopping theorem, which we briefly revisit here. Let $T$ be a stopping time of the process $X$. This means that $T \in [0, \infty]$ is a random time that is uniquely determined by the process $X$ and obeys causality, i.e., the stopping criterion that determines the stopping time $T$ is independent of the part of the trajectory of $X$ that takes place after the stopping

time $T$. Doob's optional stopping theorem states that when the stopping time $T$ is finite with probability one, and there exists a constant $c \in \mathbb{R}^+$ such that $|M(t)| < c$ for all $t \le T$, then (see, amongst others, Theorem 3.6 in [28], Corollary 2 in [24], or Theorem 3.3 in [41])

$$\langle M(T)|X(0)\rangle_{\mathbb{P}} = M(0). \tag{60}$$

Radon-Nikodym derivative process are examples of martingales, as follows readily from their definition, see Appendix A. Consequently, according to Eq. (56), the exponentiated negative entropy production is a martingale [22–24, 46]. The martingality of $e^{-S}$ is an interesting finding for physics as it can be used to constrain the fluctuations of $S(t)$. For example, using the martingale property of $e^{-S}$ together with Doob's optional stopping theorem, Refs. [22–24] derive universal laws for entropy production at stopping times $T$, including the infimum law Eq. (1). In the present paper, we use martingales to determine the statistics of infima of edge currents, extending the applicability of martingales to currents that are not the entropy production.

## 5 Martingales associated with edge currents

Following Refs. [32, 33], we identify a martingale process $M_{x \to y}$ associated with the edge current $J_{x \to y}$, which exists whenever the Markov jump process obtained by removing the edge $x \to y$ from the original process $(X, \mathbb{P}_{ss})$ has a unique stationary probability distribution.

Consider the Markov jump process $(X, \mathbb{Q}_{q^*})$ with rates

$$\ell_{u \to v} = \begin{cases} k_{u \to v}, & \text{for } (u,v) \in \{(x,y),(y,x)\}, \\ \frac{p_{ss}^{x,y}(v)}{p_{ss}^{x,y}(u)} k_{v \to u}, & \text{for } (u,v) \in \mathcal{X}^2 \setminus \{(x,y),(y,x)\}, \end{cases} \tag{61}$$

where $p_{ss}^{x,y}$ solves the equations

$$\sum_{v \in \mathcal{X}; v \neq u}' k_{u \to v} = \sum_{v \in \mathcal{X}; v \neq u}' k_{v \to u} \frac{p_{ss}^{x,y}(v)}{p_{ss}^{x,y}(u)}, \tag{62}$$

and $1 = \sum_{v \in \mathcal{X}} p_{ss}^{x,y}(v)$; the prime on the sums in Eq. (62) means that $(v,u) \notin \{(x,y),(y,x)\}$.

Equation (62) implies that the exit rates $r_p$ and $r_q$, as defined in Eq. (48), satisfy Eq. (49), and Eq. (61) implies that Eq. (51) is satisfied. Consequently, Eq. (52) applies. Setting $p^* = q^* = q_{ss}$, the latter being the stationary state of the process $(X, \mathbb{Q}_{q^*})$ (which is nota bene different from $p_{ss}^{x,y}$ and $p_{ss}$), Eq. (52) reads

$$\begin{aligned} \frac{d\mathbb{Q}_{ss}[X_0^t]}{d\mathbb{P}_{q_{ss}}[X_0^t]} &= \left( \frac{p_{ss}^{x,y}(x)k_{x \to y}}{p_{ss}^{x,y}(y)k_{y \to x}} \right)^{-J_{y \to x}(t)} \prod_{(u,v) \in \mathcal{E}; u \neq v} \left( \frac{p_{ss}^{x,y}(v)k_{v \to u}}{p_{ss}^{x,y}(u)k_{u \to v}} \right)^{\frac{J_{u \to v}(t)}{2}} \\ &= \frac{p_{ss}^{x,y}(X(t))p_{ss}(X(0))}{p_{ss}^{x,y}(X(0))p_{ss}(X(t))} \left( \frac{p_{ss}^{x,y}(x)k_{x \to y}}{p_{ss}^{x,y}(y)k_{y \to x}} \right)^{-J_{y \to x}(t)} \frac{d\mathbb{P}_{ss}[\Theta(X_0^t)]}{d\mathbb{P}_{ss}[X_0^t]}, \end{aligned} \tag{63}$$

where $\mathcal{E}$ is the set of permissible transitions, see Eq. (35), and we denote $\mathbb{Q}_{q_{ss}} = \mathbb{Q}_{ss}$ and $\mathbb{P}_{p_{ss}} = \mathbb{P}_{ss}$, as before. Equation (63) is equivalent to

$$\frac{d\mathbb{Q}_{ss}[X_0^t]}{d\mathbb{P}_{ss}[\Theta(X_0^t)]} = \frac{p_{ss}^{x,y}(X(t))q_{ss}(X(0))}{p_{ss}^{x,y}(X(0))p_{ss}(X(t))} \left( \frac{p_{ss}^{x,y}(x)k_{x \to y}}{p_{ss}^{x,y}(y)k_{y \to x}} \right)^{-J_{y \to x}(t)}. \tag{64}$$

Lastly, setting $X_0^t \to \Theta(X_0^t)$, we obtain the $\mathbb{P}_{ss}$-martingale

$$M_{x \to y}(t) := \frac{d\mathbb{Q}_{ss}[\Theta(X_0^t)]}{d\mathbb{P}_{ss}[X_0^t]} = \frac{p_{ss}^{x,y}(X(0))q_{ss}(X(t))}{p_{ss}^{x,y}(X(t))p_{ss}(X(0))} \left( \frac{p_{ss}^{x,y}(x)k_{x \to y}}{p_{ss}^{x,y}(y)k_{y \to x}} \right)^{-J_{x \to y}(t)}, \quad (65)$$

associated with the edge current $J_{x \to y}$.

Importantly, $M_{x \to y}(t)$ is a martingale because it is a Radon-Nikodym derivative process, and the latter are martingales (see Appendix A). The fact that $M_{x \to y}(t)$ is a Radon-Nikodym derivative can also be shown directly through the identity

$$M_{x \to y} = \frac{d\mathbb{R}_{ss}[X_0^t]}{d\mathbb{P}_{ss}[X_0^t]}, \quad (66)$$

where $(X, \mathbb{R}_{ss})$ is the Markov jump process with rates

$$m_{u \to v} = \begin{cases} \frac{q_{ss}(v)}{q_{ss}(u)} k_{v \to u}, & \text{for } (u,v) \in \{(x,y),(y,x)\}, \\ \frac{q_{ss}(v)}{q_{ss}(u)} \frac{p_{ss}^{x,y}(u)}{p_{ss}^{x,y}(v)} k_{u \to v}, & \text{for } (u,v) \in \mathcal{E} \setminus \{(x,y),(y,x)\}, \end{cases} \quad (67)$$

and initial distribution $p_{X(0)}(x) = q_{ss}(x)$; notice that $\mathbb{R}_{ss} = \mathbb{Q}_{ss} \circ \Theta$. At present, the identification of (65) with (66) is an insightful exercise to convince ourselves further that $M_{x \to y}$ is a martingale, and which for completeness we present in Appendix B.

Identifying in Eq. (65) the effective microscopic affinity

$$a_{x \to y}^* := \ln \frac{p_{ss}^{x,y}(x)k_{x \to y}}{p_{ss}^{x,y}(y)k_{y \to x}}, \quad (68)$$

as introduced in Ref. [32], we obtain for the $\mathbb{P}_{ss}$-martingale $M_{x \to y}(t)$ the expression

$$M_{x \to y}(t) = \frac{p_{ss}^{x,y}(X(0))q_{ss}(X(t))}{p_{ss}^{x,y}(X(t))p_{ss}(X(0))} e^{-a_{x \to y}^* J_{x \to y}(t)}. \quad (69)$$

Notice that the effective microscopic affinity $a_{x \to y}^*$, defined in Eq. (68), has a similar form as the microscopic affinity, defined in Eq. (40), with the difference that it considers the stationary probability mass function $p_{ss}^{x,y}$ in the modified process for which the transitions from $x$ to $y$, and vice versa, have been removed. Although, in general, microscopic affinities are not simply related to the "true", macroscopic affinities, for unicyclic systems it holds that the effective affinity equals the macroscopic affinity, while the microscopic affinity captures in this case a possibly small portion of the macroscopic affinity (see also Appendix C).

The effective affinity $a_{x \to y}^*$ has a kinematic meaning. Indeed, as we show in Appendix D, the martingality of $M_{x \to y}(t)$ implies that

$$a_{x \to y}^* \langle J_{x \to y}(t) \rangle_{ss} \geq 0, \quad (70)$$

where the equality holds when $a_{x \to y}^* = \langle J_{x \to y}(t) \rangle_{ss} = 0$.

In the next section, we use the martingale $M_{x \to y}$ to determine the statistics of infima of $J_{x \to y}$.

# 6 Statistics of infima of edge currents

We determine the probability mass function of $J_{x \to y}^{\inf}$ for currents with $\langle J_{x \to y}(t) \rangle_{ss} > 0$. Note that in this case, according to Eq. (70), also $a_{x \to y}^* > 0$.

In Secs. 6.1 and 6.2, we derive the probability mass functions of $J_{x \to y}^{\inf}$ for an initial state $X(0)$ that equals the source node $x$ of the edge $x \to y$ and for general initial conditions, respectively. In Secs. 6.3 and 6.4, we discuss two interesting limiting cases, namely, the case of stalled currents and the case of processes for which $(X, \mathbb{P}_{ss}^{x,y})$ obeys detailed balance, respectively.

## 6.1 Infimum law when the initial state equals the source node of the edge

First, we determine the statistics of $J^{\text{inf}}_{x \to y}$ when the initial condition $X(0) = x$. In this case, the derivations simplify.

Consider the stopping problem

$$T := \inf\left\{t \geq 0 : J_{x \to y}(t) \notin (-\ell_-, \ell_+)\right\}, \quad \text{with} \quad \ell_-, \ell_+ \in \mathbb{N} \cup \{0\}, \tag{71}$$

and where we use the convention that $T = \infty$ if $J_{x \to y}(t) \in (-\ell_-, \ell_+)$ for all times $t \geq 0$. Notice that since the edge current $J_{x \to y}(t)$ is an integer-valued stochastic process, it takes the values

$$J_{x \to y}(T) \in \{-\ell_-, \ell_+\} \tag{72}$$

at the stopping time $T$. For threshold values $\ell_+ \neq 0$, we have

$$X(T) = \begin{cases} x, & \text{if} \quad J_{x \to y}(T) = -\ell_-, \\ y, & \text{if} \quad J_{x \to y}(T) = \ell_+. \end{cases} \tag{73}$$

On the other hand, if $\ell_+ = 0$, then $T = 0$ and $X(T) = X(0) = x$.

Applying Eq. (60) from Doob's optional stopping theorem to the martingale $M_{x \to y}$ with initial condition $X(0) = x$, we obtain

$$\frac{p^{x,y}_{\text{ss}}(x)}{p_{\text{ss}}(x)}\left\langle \frac{q_{\text{ss}}(X(T))}{p^{x,y}_{\text{ss}}(X(T))} e^{-a^*_{x \to y} J_{x \to y}(T)}\Big| X(0) = x\right\rangle_{\text{ss}} = \frac{q_{\text{ss}}(x)}{p_{\text{ss}}(x)}, \tag{74}$$

where we have used that $M_{x \to y}(0) = q_{\text{ss}}(X(0))/p_{\text{ss}}(X(0))$. For the present setup,

$$\mathbb{P}_{\text{ss}}\left(T < \infty | X(0) = x\right) = 1, \tag{75}$$

and therefore for $\ell_+ \neq 0$ Eq. (74) reads

$$\begin{aligned} p_- \frac{q_{\text{ss}}(x)}{p_{\text{ss}}(x)} &\left\langle e^{-a^*_{x \to y} J_{x \to y}(T)} | J(T) = -\ell_-, X(0) = x\right\rangle_{\text{ss}} \\ &+ p_+ \frac{p^{x,y}_{\text{ss}}(x) q_{\text{ss}}(y)}{p^{x,y}_{\text{ss}}(y) p_{\text{ss}}(x)}\left\langle e^{-a^*_{x \to y} J_{x \to y}(T)} | J(T) = \ell_+, X(0) = x\right\rangle_{\text{ss}} = \frac{q_{\text{ss}}(x)}{p_{\text{ss}}(x)}, \end{aligned} \tag{76}$$

where $p_+$ and $p_-$ are the so-called splitting probabilities given by

$$p_+ = \mathbb{P}_{\text{ss}}\left(J_{x \to y}(T) = \ell_+ | X(0) = x\right), \tag{77}$$

and

$$p_- = \mathbb{P}_{\text{ss}}\left(J_{x \to y}(T) = -\ell_- | X(0) = x\right). \tag{78}$$

In the limit $\ell_+ \gg 1$, the second term on the left-hand side of the Eq. (76) converges to zero, as by assumption $\langle J_{x \to y}(t)\rangle_{\text{ss}} > 0$ and thus also $a^*_{x \to y} > 0$, and therefore

$$p_- = \left\langle e^{-a^*_{x \to y} J_{x \to y}(T)} | J(T) = -\ell_-, X(0) = x\right\rangle^{-1}_{\text{ss}} = e^{-\ell_- a^*_{x \to y}}, \tag{79}$$

with $\ell_- \in \mathbb{N} \cup \{0\}$. Since in the limit of $\ell_+ \to \infty$ it holds that

$$p_- = \mathbb{P}_{\text{ss}}(J^{\text{inf}}_{x \to y} \leq -\ell_- | X(0) = x), \tag{80}$$

Eq. (79) is the cumulative distribution of $J^{\text{inf}}_{x \to y}$. Consequently, its probability mass function reads

$$p_{J^{\text{inf}}_{x \to y}}(-\ell | X(0) = x) = e^{-\ell a^*_{x \to y}}(1 - e^{-a^*_{x \to y}}), \quad \forall \ell \in \mathbb{N} \cup \{0\}, \tag{81}$$

with mean value

$$\langle J_{x\to y}^{\text{inf}}|X(0)=x\rangle_{\text{ss}} = -\frac{e^{-a_{x\to y}^*}}{1-e^{-a_{x\to y}^*}}. \tag{82}$$

When $\langle J_{x\to y}\rangle_{\text{ss}} < 0$, we obtain the analogous result

$$p_{J_{x\to y}^{\text{sup}}}(\ell|X(0)=x) = e^{\ell a_{x\to y}^*}\left(1-e^{a_{x\to y}^*}\right), \quad \forall \ell \in \mathbb{N} \cup \{0\}, \tag{83}$$

with the mean value

$$\langle J_{x\to y}^{\text{sup}}|X(0)=x\rangle_{\text{ss}} = \frac{e^{a_{x\to y}^*}}{1-e^{a_{x\to y}^*}}. \tag{84}$$

## 6.2 Infimum law for general initial conditions

We follow a derivation similar to the one presented in the previous section, but now for general initial conditions $X(0)=x_0$.

Applying Eq. (60) from Doob's optional stopping theorem to the martingale $M_{x\to y}$, given by Eq. (69), we obtain

$$\frac{p_{\text{ss}}^{x,y}(x_0)}{p_{\text{ss}}(x_0)}\left\langle\frac{q_{\text{ss}}(X(T))}{p_{\text{ss}}^{x,y}(X(T))}e^{-a_{x\to y}^* J_{x\to y}(T)}|X(0)=x_0\right\rangle_{\text{ss}} = \frac{q_{\text{ss}}(x_0)}{p_{\text{ss}}(x_0)}. \tag{85}$$

Following similar steps as those leading to Eq. (79), we obtain in the limit of $\ell_+ \gg 1$,

$$\mathbb{P}_{\text{ss}}(J_{x\to y}^{\text{inf}} \le -\ell_-|X(0)=x_0) = \frac{q_{\text{ss}}(x_0)}{q_{\text{ss}}(x)}\frac{p_{\text{ss}}^{x,y}(x)}{p_{\text{ss}}^{x,y}(x_0)}e^{-a_{x\to y}^*\ell_-}, \quad \forall \ell_- \in \mathbb{N}, \tag{86}$$

and

$$\mathbb{P}_{\text{ss}}(J_{x\to y}^{\text{inf}} \le 0|X(0)=x_0) = 1. \tag{87}$$

Notice that for general initial conditions, Eq. (73) holds for values $\ell_- \ne 0$ and $\ell_+ \ne 0$, but not when either of the two thresholds is zero, and therefore $\ell_- \ne 0$ in Eq. (86). From Eqs. (86) and (87) follows the probability mass function of $J_{x\to y}^{\text{inf}}$, which for $\ell \in \mathbb{N}$ is given by

$$p_{J_{x\to y}^{\text{inf}}}(-\ell|X(0)=x_0) = \frac{q_{\text{ss}}(x_0)}{q_{\text{ss}}(x)}\frac{p_{\text{ss}}^{x,y}(x)}{p_{\text{ss}}^{x,y}(x_0)}e^{-a_{x\to y}^*\ell}\left(1-e^{-a_{x\to y}^*}\right), \tag{88}$$

and for $\ell = 0$ it is

$$p_{J_{x\to y}^{\text{inf}}}(0|X(0)=x_0) = 1 - \frac{q_{\text{ss}}(x_0)}{q_{\text{ss}}(x)}\frac{p_{\text{ss}}^{x,y}(x)}{p_{\text{ss}}^{x,y}(x_0)}e^{-a_{x\to y}^*}. \tag{89}$$

Equations (88-89) readily imply Eq. (14) for the probability mass function of the infimum and Eq. (15) for the average value of the infimum, where we identified

$$p_{\text{esc}}(x_0) = p_{J_{x\to y}^{\text{inf}}}(0|X(0)=x_0). \tag{90}$$

For $x_0 = x$, $p_{\text{esc}}$ is given by Eq. (16).

## 6.3 Stalled currents

We say that a current is stalled when

$$\langle J_{x\to y}(t)\rangle_{\mathrm{ss}} = 0. \tag{91}$$

Notice that stalled currents may exist in nonequilibrium stationary states.

For nearly stalled currents $a^*_{x\to y} \approx 0$ (as shown in Appendix D and in Refs. [32, 33]), $p_{\mathrm{esc}}(x_0) \approx 0$, and $\bar{j}_{x\to y} \approx 0$. Consequently, in this limiting case the Eq. (14) becomes the exponential distribution

$$p_{J^{\inf}_{x\to y}}(-\ell|X(0) = x_0) = a^*_{x\to y} e^{-\ell a^*_{x\to y}}, \quad \ell \in \mathbb{R}^+, \tag{92}$$

with mean

$$\langle J^{\inf}_{x\to y}|X(0) = x_0\rangle_{\mathrm{ss}} = -\frac{1}{a^*_{x\to y}}. \tag{93}$$

Hence, the mean of the infimum diverges in the vicinity of a stalling point.

Equation (92) implies that the statistics of infima of edge currents $J_{x\to y}$ in the vicinity of nonequilibrium stalled states are identical to those in the vicinity of equilibrium states. Therefore, a marginal observer that only observes the edge current $J_{x\to y}$ cannot make a distinction between a nonequilibrium stalled state and an equilibrium state from the measurements of the extreme values of $J_{x\to y}$ only.

## 6.4 Markov processes driven out of equilibrium by a single edge

Although Ref. [1] derives an explicit expression for the effective rates $a^*_{x\to y}$ in terms of the minors of the rate matrix, this leads, in general, to a long expression without clear physical interpretation. Here we consider a limiting case for which $a^*_{x\to y}$ admits a simple, explicit expression.

We consider Markov processes $(X, \mathbb{P}_{\mathrm{ss}})$ for which the process $(X, \mathbb{P}^{x,y}_{\mathrm{ss}})$ satisfies detailed balance, where $\mathbb{P}^{x,y}_{\mathrm{ss}}$ is the measure of the stationary Markov jump process obtained by removing the edges $x \to y$ and $y \to x$ from the original process. Hence, in this case the stationary distribution $p^{x,y}_{\mathrm{ss}} = p_{\mathrm{eq}}$ is an equilibrium distribution.

Consequently, the effective affinity $a^*_{x\to y}$ takes the form

$$a^*_{x\to y} = \ln\frac{p_{\mathrm{eq}}(x)}{p_{\mathrm{eq}}(y)} + \ln\frac{k_{x\to y}}{k_{y\to x}}. \tag{94}$$

Parameterising

$$k_{x\to y} = \omega_{x,y}\, p_{\mathrm{eq}}(y)\, e^{\frac{f_{x\to y}}{2\mathsf{T}_{\mathrm{env}}}}, \tag{95}$$

where $\omega_{x,y} = \omega_{y,x}$ is a symmetric kinetic parameter and $f_{x\to y} = -f_{x\to y}$ a thermodynamic force, we obtain

$$a^*_{x\to y} = \frac{f_{x\to y}}{\mathsf{T}_{\mathrm{env}}}. \tag{96}$$

Hence, in the present case, the microscopic affinity is directly related to the thermodynamic force $f_{x\to y}$.

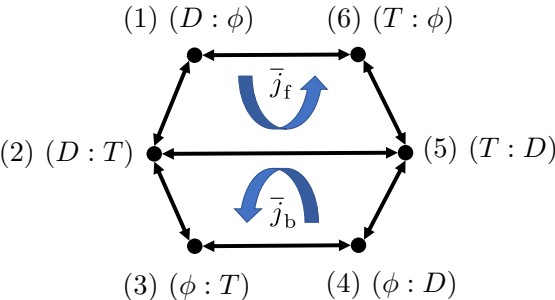

Figure 2: Six-state model for molecular motors with two motor heads [figure taken from [19]].

# 7 Illustration of infimum laws for two-headed molecular motors

We use a model for two-headed molecular motors to illustrate the implications of the infimum laws for edge currents on the dynamics of a physical process. We first introduce in Sec. 7.1 a Markov jump process for molecular motor dynamics, and subsequently in Sec. 7.2, we use this model to study the extreme values in the position of molecular motors.

## 7.1 Model for two-headed molecular motors

Consider a molecular motor bound to a one-dimensional substrate that walks with discrete steps of size $\pm\delta$. The molecular motor is driven out of equilibrium by two thermodynamic forces, namely, an input of free energy $\Delta\mu$ – due to the hydrolysis of adenosine triphosphate (ATP) into adenosine diphosphate (ADP) and an inorganic phosphate (P) — and a mechanical force $f_{\mathrm{mech}}$. The average entropy production rate is

$$\dot{s} = \frac{\langle S(t)\rangle_{\mathrm{ss}}}{t} = \frac{\Delta\mu}{\mathsf{T}_{\mathrm{env}}}\bar{j}_{\mathrm{fuel}} - \frac{f_{\mathrm{mech}}\delta}{\mathsf{T}_{\mathrm{env}}}\bar{j}_{\mathrm{pos}}\,, \tag{97}$$

where $\mathsf{T}_{\mathrm{env}}$ is the temperature of the environment, $\bar{j}_{\mathrm{fuel}}$ is the average rate of the reaction ATP$\rightarrow$ ADP+P minus the rate of the reverse reaction ADP+P$\rightarrow$ ATP, and $\bar{j}_{\mathrm{pos}}$ is the average rate at which the motor moves forwards minus the rate at which the motor moves backwards. The minus sign in front of $f_{\mathrm{mech}}$ indicates that the mechanical force pushes the motor in the negative direction. The free energy associated with the hydrolysis reaction is given by

$$\Delta\mu = \mathsf{T}_{\mathrm{env}}\ln\left(K_{\mathrm{eq}}\frac{[\mathrm{ATP}]}{[\mathrm{ADP}][\mathrm{P}]}\right)\,, \tag{98}$$

where $K_{\mathrm{eq}}$ is the equilibrium constant of the hydrolysis interaction, and [ATP], [ADP] and [P] are, respectively, the concentrations of ATP, ADP and P in the surrounding medium.

In what follows, we consider the six-state model for two-headed molecular motors as introduced in Ref. [36], which is a Markov jump process that describes the basic features of the molecular motor's thermodynamics as described by Eq. (97). In this model, the position of the molecular motor along the biofilament is proportional to an edge current, and hence the theory for extreme values of Sec. 6 applies.

The six states represent the different chemical states of the rear and front motor heads, both of which can be in an ATP-bound state (T), ADP-bound state (ADP) and nucleotide-free

state ($\phi$). Since the motor heads move out of phase, the three states ($\phi : \phi$), ($D : D$), and ($T : T$) are excluded, and the process takes six possible states,

$$X(t) \in \{(D : \phi), (T : \phi), (T : D), (\phi : D), (\phi : T), (D : T)\} . \tag{99}$$

For convenience, we also label states by 1 to 6, as indicated in Fig. 2. The pairs of states ($D : \phi$) and ($\phi : D$) — but also ($D : T$) and ($T : D$), or ($T : \phi$) and ($\phi : T$) — are not identical, as in the state ($D : \phi$) the rear motor head is bound to ADP and the front motor head is in the nucleotide free state, while in ($\phi : D$) it is the other way around. The asymmetry in the configurations ($D : \phi$) and ($\phi : D$) is due to an asymmetry in the periodic, electric potential of the one-dimensional substrate to which the motor is bound.

The dynamics of $X(t)$ is governed by a Markov jump process with the nonzero transition rates indicated by arrows in Fig. 2. All transitions, except those between ($D : T$) and ($T : D$), are chemical transitions. On the other hand, the transition from ($D : T$) to ($T : D$), and vice-versa, is a mechanical transition where the motor heads swap position. Therefore, the position $J_{\text{pos}}(t)$ of the motor is the edge current

$$J_{\text{pos}}(t) := J_{(T:D)\to(D:T)}(t) = J_{2\to5}(t), \tag{100}$$

where consistently with the setup of Sec. 4, we have set $J_{\text{pos}}(0) = 0$.

Following Refs. [36,37], we parameterise the jump rates corresponding to the mechanical transitions as

$$k_{2\to5}(f_{\text{mech}}) = k_{2\to5}(0)e^{-\theta \frac{f_{\text{mech}}\delta}{T_{\text{env}}}}, \tag{101}$$

and

$$k_{5\to2}(f_{\text{mech}}) = k_{5\to2}(0)e^{(1-\theta)\frac{f_{\text{mech}}\delta}{T_{\text{env}}}}, \tag{102}$$

and the chemical transitions are parameterised as

$$k_{i\to j}(f_{\text{mech}}) = \frac{2k_{i\to j}(0)}{1 + e^{\chi_{ij}\frac{f_{\text{mech}}\delta}{T_{\text{env}}}}}, \tag{103}$$

with $\chi_{ij} = \chi_{ji}$. Note that also the chemical transitions depend on the mechanical force $f_{\text{mech}}$, as the force will deform the motor heads affecting the rate of chemical reactions. However, since $\chi_{ij} = \chi_{ji}$, it holds that

$$\frac{k_{i\to j}(f_{\text{mech}})}{k_{j\to i}(f_{\text{mech}})} = \frac{k_{i\to j}(0)}{k_{j\to i}(0)}, \tag{104}$$

for $(i, j) \notin \{(2,5), (5,2)\}$ and all values of $f_{\text{mech}}$, and consequently $f_{\text{mech}}$ provides a nonzero contribution to the microscopic affinity $a_{2\to5}$ only.

The concentrations of [ADP] and [P] are assumed to be constant, and the dependence on [ATP] enters into the model through

$$k_{1\to2}(0) = k_{1\to2}^{\text{bi}}[\text{ATP}], \tag{105}$$

and

$$k_{4\to5}(0) = k_{4\to5}^{\text{bi}}[\text{ATP}], \tag{106}$$

where the $k_{1\to2}^{\text{bi}}$ and $k_{4\to5}^{\text{bi}}$ are rate constants whose value depend on the properties of the motor. Due to the equivalence of transitions in the backward and forward cycles, we set $k_{3\to2}(0) = k_{6\to5}(0)$, $k_{2\to3}(0) = k_{5\to6}(0)$, $k_{3\to4}(0) = k_{6\to1}(0)$, $k_{4\to3}(0) = k_{1\to6}(0)$, $k_{4\to5}(0) = k_{1\to2}(0)$, $\chi_{23} = \chi_{56}$, $\chi_{34} = \chi_{61}$, and $\chi_{45} = \chi_{12}$. In addition, following Ref. [19], we use

$$k_{5\to2}(0) = k_{2\to5}(0)\sqrt{\frac{k_{5\to4}(0)}{k_{2\to1}(0)}}, \tag{107}$$

so that the six-state model satisfies detailed balance when

$$f_{\text{mech}} = 0 \quad \text{and} \quad [\text{ATP}] = \frac{k_{1\to6}(0)}{k_{6\to1}(0)} \frac{k_{6\to5}(0)}{k_{5\to6}(0)} \frac{\sqrt{k_{5\to4}(0)k_{2\to1}(0)}}{k^{\text{bi}}_{4\to5}}, \tag{108}$$

and accordingly both thermodynamic forces $f_{\text{mech}} = \Delta\mu = 0$ at these values of the control parameters. The stationary distribution $p_{\text{ss}} = p_{\text{eq}}$ at equilibrium is presented in Appendix E. The remaining constants $k_{i\to j}(0)$, $\chi_{ij}$, and $\theta$ can be determined by fitting the model to single molecule motility data, and for Kinesin-1 we report these values in Appendix F.

As shown in Fig. 2, the model has three cycles, one corresponding to a forward motion at a rate $\bar{j}_{\text{f}}$, one corresponding to a backward motion at a rate $\bar{j}_{\text{b}}$, and one for which the motor does not move but hydrolyses two ATP molecules into ADP and P at a rate $\bar{j}_0$. The corresponding thermodynamic affinities are

$$\frac{a_{\text{f}}}{\mathsf{T}_{\text{env}}} = \ln \frac{k_{1\to2}k_{2\to5}k_{5\to6}k_{6\to1}}{k_{1\to6}k_{6\to5}k_{5\to2}k_{2\to1}} = \frac{\Delta\mu}{\mathsf{T}_{\text{env}}} - \frac{f_{\text{mech}}\delta}{\mathsf{T}_{\text{env}}}, \tag{109}$$

$$\frac{a_{\text{b}}}{\mathsf{T}_{\text{env}}} = \ln \frac{k_{2\to3}k_{3\to4}k_{4\to5}k_{5\to2}}{k_{2\to5}k_{5\to4}k_{4\to3}k_{3\to2}} = \frac{\Delta\mu}{\mathsf{T}_{\text{env}}} + \frac{f_{\text{mech}}\delta}{\mathsf{T}_{\text{env}}}, \tag{110}$$

and

$$\frac{a_0}{\mathsf{T}_{\text{env}}} = \ln \frac{k_{2\to3}k_{3\to4}k_{4\to5}k_{5\to6}k_{6\to1}k_{1\to2}}{k_{3\to2}k_{2\to1}k_{1\to6}k_{6\to5}k_{5\to4}k_{4\to3}} = \frac{2\Delta\mu}{\mathsf{T}_{\text{env}}}. \tag{111}$$

The total, average rate of dissipation, as defined in Eq. (58), is thus given by

$$\dot{s} = \bar{j}_{\text{f}}\frac{a_{\text{f}}}{\mathsf{T}_{\text{env}}} + \bar{j}_{\text{b}}\frac{a_{\text{b}}}{\mathsf{T}_{\text{env}}} + \frac{a_0}{\mathsf{T}_{\text{env}}}\bar{j}_0, \tag{112}$$

which provides an alternative decomposition of the average entropy production rate from Eq. (97).

## 7.2 Infimum laws for the position of molecular motors

We determine the statistics of the infima $J^{\text{inf}}_{\text{pos}}$ in the position $J_{\text{pos}}$ of molecular motors, as described by the six state model illustrated in Fig. 2. Since $J_{\text{pos}}$ is the edge current corresponding to the $2 \to 5$ transition, see Eq. (100), this boils down, according to Eqs. (14) and (15), to evaluating the effective affinity

$$a^*_{2\to5} = \ln \frac{p^{2,5}_{\text{ss}}(2)}{p^{2,5}_{\text{ss}}(5)} + \ln \frac{p_{\text{eq}}(5)}{p_{\text{eq}}(2)} - \frac{f_{\text{mech}}\delta}{\mathsf{T}_{\text{env}}}. \tag{113}$$

To derive (113), we have used the expression (68) for the effective affinity together with the detailed balance condition

$$\frac{k_{2\to5}(0)}{k_{5\to2}(0)} = \frac{p_{\text{eq}}(5)}{p_{\text{eq}}(2)}, \tag{114}$$

satisfied by the rates $k_{2\to5}(0)$ and $k_{5\to2}(0)$. The distribution $p_{\text{eq}}(x)$ is the stationary distribution at equilibrium conditions $\Delta\mu = f_{\text{mech}} = 0$, and the explicit values of $p_{\text{eq}}$ are presented in Appendix E.

In what follows, we determine $a^*_{2\to5}$ in three cases: (i) chemical equilibrium but mechanical driving ($\Delta\mu = 0, |f_{\text{mech}}| > 0$); (ii) mechanical equilibrium but chemical driving ($|\Delta\mu| > 0, f_{\text{mech}} = 0$); (iii) mechanical and chemical driving ($|\Delta\mu| > 0, |f_{\text{mech}}| > 0$). In cases (i) and (ii) the stalled state is the equilibrium state, while in the latter the stalled state is, in general, a nonequilibrium state.

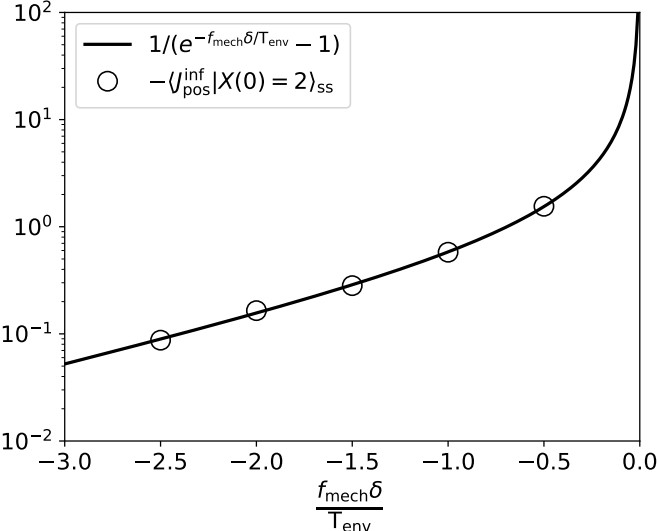

Figure 3: Mean infimum for the position $J_{pos}(t) = J_{2 \to 5}(t)$ of a molecular motor in the model of Sec. 7.1 (also illustrated in Fig. 2) as a function of the mechanical force $f_{mech}$ and without chemical driving, i.e., $\Delta\mu = 0$. Theoretical results given by Eq. (116) (line) are compared with empirical averages from numerical simulations (markers). Simulation results are empirical averages for the most negative value of the position of the molecular motor averaged over $5e+3$ realisations of the process when the initial state $X(0) = 2$.

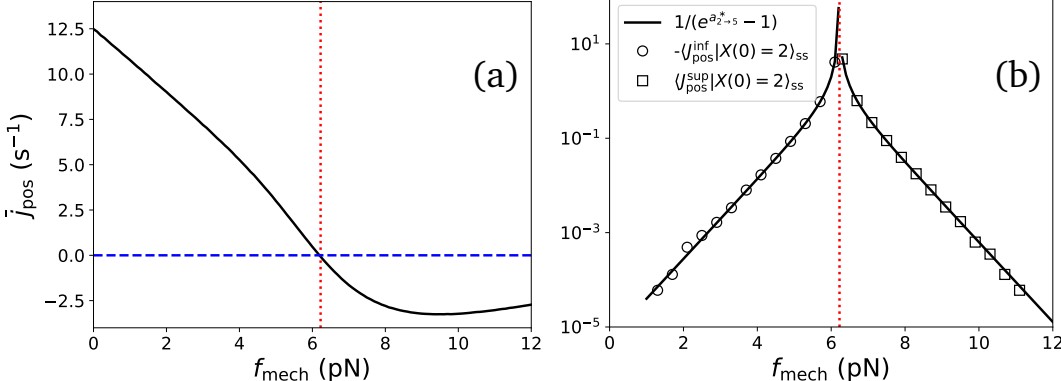

Figure 4: Illustration of the infimum law on the position $J_{pos}(t) = J_{2 \to 5}(t)$ of the molecular motor model for Kinesin-1 illusrated in Fig. 2. Parameters used are as described in Sec. 7 with $[ATP] = 10\mu M$. (a) Number of molecular motor steps per second $\bar{j}_{pos}$ as a function of the mechanical force $f_{mech}$; (b) mean infimum $\langle J_{pos}^{inf}|X(0) = 2\rangle_{ss}$ ($f_{mech} < f_s$) or supremum $\langle J_{pos}^{sup}|X(0) = 2\rangle_{ss}$ ($f_{mech} > f_s$) as a function of the mechanical force $f_{mech}$, where $f_s \approx 6.2$pN is the stalling force. Theoretical curves (lines), obtained from plotting the Eq. (15), are compared with results from continuous-time Monte-Carlo simulations (markers); each marker is the sample average over $10^5$ realisations of the process. The stalling force is denoted by a vertical dotted line.

### 7.2.1 Chemical equilibrium and mechanical driving

We consider a molecular motor in chemical equilibrium with its environment — [ATP] is given by Eq. (108), such that $\Delta\mu = 0$ — and that is driven out of equilibrium by a mechanical force $f_{\mathrm{mech}}$. The limiting case of Sec. 6.4 applies here, i.e., $p_{\mathrm{ss}}^{(2,5)} = p_{\mathrm{eq}}$, as $\chi_{ij} = \chi_{ji}$, and hence the force $f_{\mathrm{mech}}$ does not change the ratios of the transition rates, except for the transition from 2 to 5, and vice versa. Consequently, the effective affinity is determined by the thermodynamic force through

$$a_{2\to5}^* = -\frac{f_{\mathrm{mech}}\delta}{\mathsf{T}_{\mathrm{env}}}\,. \tag{115}$$

Using Eq. (115) in Eqs. (15) and (84), we obtain explicit analytical expressions for the mean extreme values of the molecular motor position, viz.,

$$\langle J_{\mathrm{pos}}^{\inf}|X(0)=2\rangle_{\mathrm{ss}} = -\frac{1}{e^{-\frac{f_{\mathrm{mech}}\delta}{\mathsf{T}_{\mathrm{env}}}} - 1}\,, \quad\text{and}\quad \langle J_{\mathrm{pos}}^{\sup}|X(0)=2\rangle_{\mathrm{ss}} = \frac{1}{e^{\frac{f_{\mathrm{mech}}\delta}{\mathsf{T}_{\mathrm{env}}}} - 1}\,, \tag{116}$$

for $f_{\mathrm{mech}} < 0$ and $f_{\mathrm{mech}} > 0$, respectively.

In Fig. 3, we plot Eq. (116) as a function of $f_{\mathrm{mech}}$ together with results from continuous-time Monte Carlo simulations obtained from empirical averages over $10^4$ trajectories. Notice the divergence of the mean value of the infimum when approaching the equilibrium state $f_{\mathrm{mech}} \to 0$.

### 7.2.2 Mechanical equilibrium and chemical driving

Now, we consider the opposite case for which the motor is driven out of equilibrium by chemical fuel, while the mechanical force equals zero.

In this case, the effective affinity $a_{2\to5}^*$ does not admit a simple thermodynamic interpretation in terms of the nonequilibrium forcing $\Delta\mu$. Indeed, the thermodynamic force $\Delta\mu$ acts on two edges, namely, $1 \to 2$ and $4 \to 5$, and hence the limiting case of Sec. 6.4 does not apply here. Consequently, also Eq. (96) that expresses the effective affinity in terms of the thermodynamic force does not hold.

The fact that $a_{2\to5}^*$ does not admit a simple thermodynamic expression in terms of the thermodynamic force $\Delta\mu$ is even true when $\Delta\mu \to 0$. Indeed, in the linear response regime

$$a_{2\to5}^* = \frac{1}{p_{\mathrm{eq}}(5)k_{5\to2}(0)}\frac{\Delta\mu}{\mathsf{T}_{\mathrm{env}}}\left(k_{2\to5}(0)p_\mu^{2,5}(2) - k_{5\to2}(0)p_\mu^{2,5}(5)\right) + O\left(\left(\frac{\Delta\mu}{\mathsf{T}_{\mathrm{env}}}\right)^2\right), \tag{117}$$

which does not admit a simple interpretation in terms of the nonequilibrium forcing $\Delta\mu$ and the Onsager coefficients of the currents in the process $(X, \mathbb{P}_{\mathrm{ss}})$.

### 7.2.3 Chemical and mechanical driving

Lastly, we discuss extreme values in the vicinity of nonequilibrium, stalled states. We set $[\mathrm{ATP}] = 10\mu\mathrm{M}$, in which case the motor stalls at a force $f_{\mathrm{mech}} = f_{\mathrm{s}} \approx 6.2\mathrm{pN}$, as shown in Panel (a) of Fig. 4. The average rate of dissipation $\dot{s} > 0$, and hence this is a nonequilibrium stalled state for which the motor does not realise average motion despite constantly consuming chemical energy. Nevertheless, as shown in Panel (b) of Fig. 4, the mean value of the extreme value of $J_{\mathrm{pos}}$ diverges near $f_{\mathrm{mech}} = f_{\mathrm{s}}$, similar to the mean extreme value near equilibrium shown in Fig. 3. This follows from the fact that the statistics of infima, as determined by Eqs. (92) and (93), are the same for stalled states and for equilibrium states.

# 8 Estimating the average entropy production rate based on the extreme value statistics of an edge current

Given that average entropy production rates can be estimated from the fluctuations of currents at a fixed time, see Ref. [47], it is natural to expect that average entropy production rates can also be estimated with the extreme value statistics of currents. In this Section, we introduce two estimators for dissipation based on extreme value statistics of currents, namely, $\hat{s}_{\text{inf}}$ that applies to arbitrary currents $J$, and $\hat{\mathring{s}}_{\text{inf}}$ that applies to edge currents $J_{x \to y}$. We compare the bias of the estimators $\hat{s}_{\text{inf}}$ and $\hat{\mathring{s}}_{\text{inf}}$ with estimators that have been studied previously in the literature, in particular, the thermodynamic uncertainty ratio $\hat{s}_{\text{TUR}}$ [7,9,47] and a naive estimator $\hat{s}_{\text{KL}}$ based on neglecting nonMarkovian statistics in the Kullback-Leibler divergence of the integrated edge current [48–52].

   We start with reviewing in Sec. 8.1 estimators of $\dot{s}$ that have been studied previously in the literature. Subsequently, in Sec. 8.2, we discuss the two estimators $\hat{s}_{\text{inf}}$ and $\hat{\mathring{s}}_{\text{inf}}$ that are based on extreme value statistics. Lastly, in Sec. 8.3, we evaluate the quality of the different estimators of dissipation when applied to the current $J_{\text{pos}}$ of the molecular motor model defined in Sec. 7.1.

## 8.1 Estimators of the average entropy production rate revisited

To evaluate the quality of estimators that are based on extreme value statistics, we first review three well-studied estimators of dissipation, all of which are evaluated on the trajectories $J_0^t$ of an arbitrary current $J$, as defined in Eq. (12):

1. The *Kullback-Leibler divergence $\hat{s}_{\text{KL}}$*: Let $\mathcal{J}$ be the set of jump sizes of the current $J(t)$, excluding jumps of size zero. Then, the Kullback-Leibler divergence of the current $J$, neglecting non-Markovian statistics, is defined by [48–53]

$$\hat{s}_{\text{KL}} := \sum_{j \in \mathcal{J}} \dot{n}_j \ln \frac{\dot{n}_j}{\dot{n}_{-j}}, \tag{118}$$

where $\dot{n}_j$ denotes the rate at which the current $J$ makes jumps of size $J(t) - J(t^-) = j$, and where $t^-$ denotes a time infinitesimal smaller than $t$. The estimator $\hat{s}_{\text{KL}}$ is obtained from the Kullback-Leibler divergence

$$\left\langle \ln \frac{d\mathbb{P}_{\text{ss}}[J_0^t]}{d(\mathbb{P}_{\text{ss}} \circ \Theta)[J_0^t]} \right\rangle_{\text{ss}}, \tag{119}$$

by ignoring non-Markovian statistics in the trajectory of $J_0^t$; notice that in Eq. (119) $\mathbb{P}_{\text{ss}}[J_0^t]$ denotes the probability measure $\mathbb{P}_{\text{ss}}$ constrained to the $\sigma$-algebra generated by $J_0^t$. Since time-reversal flips the sign of the current, i.e., $J(\Theta(\omega), t) = -J(\omega, t)$, we obtain in the logarithm of Eq. (118) the ratio between $n_j$ and $n_{-j}$.

The Kullback-Leibler divergence lower bounds $\dot{s}$, i.e.,

$$\hat{s}_{\text{KL}} \le \dot{s}. \tag{120}$$

However, when the statistics of the current $J$ contain strong non-Markovian effects and when $J$ is not proportional to the entropy production $S$, than $\hat{s}_{\text{KL}}$ provides a poor estimate of $\dot{s}$ as it does not capture the irreversibility in the non-Markovian statistics [48–53].

2. The *thermodynamic uncertainty ratio $\hat{s}_{\text{TUR}}$*: this ratio is defined by [7,9,47]

$$\hat{s}_{\text{TUR}} := 2 \frac{\bar{j}^2 t}{\sigma_{J(t)}^2}, \tag{121}$$

where $\bar{j} = \langle J(t)\rangle_{\mathrm{ss}}/t$ and $\sigma^2_{J(t)} = \langle J^2(t)\rangle_{\mathrm{ss}} - \langle J(t)\rangle^2_{\mathrm{ss}}$. For Markov jump processes the thermodynamic uncertainty ratio lower bounds $\dot{s}$, i.e.,

$$\hat{s}_{\mathrm{TUR}} \leq \dot{s}, \tag{122}$$

see Refs. [5–7, 9]. However, in Markov jump processes that are governed far from thermal equilibrium, $\hat{s}_{\mathrm{TUR}}/\dot{s} \approx 0$ [19], and hence the thermodynamic uncertainty ratio captures a negligible fraction of the dissipation in this limit. Notice that in overdamped Langevin processes $\hat{s}_{\mathrm{TUR}}/\dot{s} \approx 1$ for small $t$ when $J = S$, as for example shown in Ref. [54]. However, this relies on the fact that the distribution of $S(t)$ is Gaussian for small $t$, which does not apply to processes with jumps.

3. The *first-passage ratio* $\hat{s}_{\mathrm{FPR}}$: Let $T = \inf\{t \geq 0 : J(t) \notin (-\ell_-, \ell_+)\}$ be the first time a current $J$ exits an open interval $(-\ell_-, \ell_+)$, and let us assume $\langle J(t)\rangle_{\mathrm{ss}} > 0$. The fist-passage ratio of the current $J$ is defined by [10, 16, 19]

$$\hat{s}_{\mathrm{FPR}}(\ell_+, \ell_-) := \frac{\ell_+}{\ell_-} \frac{|\ln p_-|}{\langle T\rangle_{\mathrm{ss}}}, \tag{123}$$

where $p_- = \mathbb{P}_{\mathrm{ss}}(J(T) \leq -\ell_-)$ is the probability that the current goes below the threshold $-\ell_-$ before exceeding the threshold $\ell_+$. Ref. [16] shows that in the limit of large thresholds $\ell_-$ and $\ell_+$, while keeping the ratio $\ell_-/\ell_+$ fixed,

$$\hat{s}_{\mathrm{FPR}} \leq \dot{s}. \tag{124}$$

In addition, when $J$ is proportional to $S$, then in the same limit $\hat{s}_{\mathrm{FPR}} = \dot{s}$. Although results in Ref. [19] indicate that in general the bias of $\hat{s}_{\mathrm{FPR}}$ is smaller than the bias in $\hat{s}_{\mathrm{KL}}$ and $\hat{s}_{\mathrm{TUR}}$, the estimator $\hat{s}_{\mathrm{FPR}}$ has the drawback that it should be evaluated at large thresholds $\ell_-$ and $\ell_+$.

In the case of $J = J_{x\to y}$, we can, using the martingale methods discussed in this paper, evaluate the bias of the estimator $\hat{s}_{\mathrm{FPR}}$ in the limit of large thresholds. Indeed, as shown in Ref. [16],

$$\langle T\rangle_{\mathrm{ss}} = \frac{\ell_+}{\bar{j}_{x\to y}}(1 + o_{\ell_{\min}}(1)), \tag{125}$$

where $o_{\ell_{\min}}(1)$ is the little-o notation that denotes an arbitrary function that converges to zero when both $\ell_+$ and $\ell_-$ diverge while their ratio is kept fixed, and $\bar{j}_{x\to y} = \langle J_{x\to y}(t)\rangle_{\mathrm{ss}}/t$. Additionally, Eq. (14) implies that for $\bar{j}_{x\to y} > 0$,

$$\frac{|\ln p_-|}{\ell_-} = a^*_{x\to y}\left(1 + o_{\ell_{\min}}(1)\right). \tag{126}$$

Using Eqs. (14) and (126) in (123), we obtain

$$\hat{s}_{\mathrm{FPR}} = a^*_{x\to y}\bar{j}_{x\to y}\left(1 + o_{\ell_{\min}}(1)\right). \tag{127}$$

Hence, the average rate of dissipation estimated by a marginal observer is the current rate $\bar{j}_{x\to y}$ times the effective affinity $a^*_{x\to y}$, which justifies calling $a^*_{x\to y}$ an effective affinity. It follows from Eq. (130) that

$$a^*_{x\to y}\bar{j}_{x\to y} \leq \dot{s}, \tag{128}$$

which has also been derived in Ref. [33, 35] using a different approach.

## 8.2 Estimators of dissipation based on infimum statistics

We introduce two estimators of dissipation based on the infimum statistics of currents.

1. The *infimum ratio* $\hat{s}_{\mathrm{inf}}$: defined in Eq. (3), the infimum ratio applies to generic currents $J$ of the form Eq. (12). The infimum ratio is related to the first-passage ratio $\hat{s}_{\mathrm{FPR}}$ through

$$\hat{s}_{\mathrm{inf}}(\ell_-) = \lim_{\ell_+ \to \infty} \hat{s}_{\mathrm{FPR}}(\ell_-, \ell_+), \tag{129}$$

and therefore it inherits the properties of $\hat{s}_{\mathrm{FPR}}$, viz.,

$$\lim_{\ell \to \infty} \hat{s}_{\mathrm{inf}}(\ell) \leq \dot{s}, \tag{130}$$

and

$$\lim_{\ell \to \infty} \hat{s}_{\mathrm{inf}}(\ell) = \dot{s}, \tag{131}$$

when $J$ is proportional to $S$. Moreover, for currents $J$ that are edge currents $J_{x \to y}$, it follows from Eq. (127) that

$$\lim_{\ell \to \infty} \hat{s}_{\mathrm{inf}}(\ell) = a^*_{x \to y} \bar{j}_{x \to y}. \tag{132}$$

The drawback of $\hat{s}_{\mathrm{inf}}$ is that the Eqs. (130)-(132) hold asymptotically in the limit of large thresholds $\ell$. However, for edge currents we can resolve this infinite threshold problem with the next estimator that we discuss.

2. The *modified infimum ratio* $\hat{\bar{s}}_{\mathrm{inf}}$: this ratio, defined in Eq. (9), applies to edge currents $J_{x \to y}$. It follows readily from the definitions Eqs. (3) and (9) and the result Eq. (14) that

$$\hat{\bar{s}}_{\mathrm{inf}} = a^*_{x \to y} \bar{j}_{x \to y}, \tag{133}$$

and hence the modified infimum ratio at finite values of $\ell$ equals the infimum ratio $\hat{s}_{\mathrm{inf}}(\ell)$ in the limit of large $\ell$. Although the modified infimum ratio does not apply to generic currents $J$, it has the advantage that it uses $p_-(\ell)$ at finite values of $\ell$, and hence it resolves the infinite threshold problem of the estimators $\hat{s}_{\mathrm{inf}}$ and $\hat{s}_{\mathrm{FPR}}$.

## 8.3 Estimation of dissipation in a molecular motor model

We show the different estimators at work on the paradigmatic example of a molecular motor that is bound to a biofilament. We consider an experimenter that measures the position $J_{\mathrm{pos}}$ of the two-headed molecular motor, as defined in Sec. 7.1. To this aim, the experimenter uses the three estimators $\hat{s}_{\mathrm{KL}}$, $\hat{s}_{\mathrm{TUR}}$, and $\hat{\bar{s}}_{\mathrm{inf}} = \lim_{\ell \to \infty} \hat{s}_{\mathrm{inf}}(\ell)$.

Since $J_{\mathrm{pos}} = J_{2 \to 5}$, it holds that

$$\hat{\bar{s}}_{\mathrm{inf}}(\ell) = a^*_{2 \to 5} \bar{j}_{2 \to 5}, \tag{134}$$

for $\ell \in \mathbb{N}$. Using in Eq. (113) the explicit expressions for $p^{2,5}_{\mathrm{ss}}(x)$ reported in Appendix G, we obtain for the effective affinity the formula

$$\begin{aligned} a^*_{2 \to 5} = {} & \ln \frac{k_{1 \to 2} k_{2 \to 3} k_{3 \to 4} + k_{3 \to 2} k_{4 \to 3} k_{5 \to 4}}{k_{1 \to 2} k_{2 \to 3} k_{3 \to 4} + k_{2 \to 1} k_{3 \to 2} k_{4 \to 3}} \\ & + \frac{1}{2} \ln \frac{k_{2 \to 1}(0)}{k_{5 \to 4}(0)} - \frac{f_{\mathrm{mech}} \delta}{\mathrm{T}_{\mathrm{env}}}, \end{aligned} \tag{135}$$

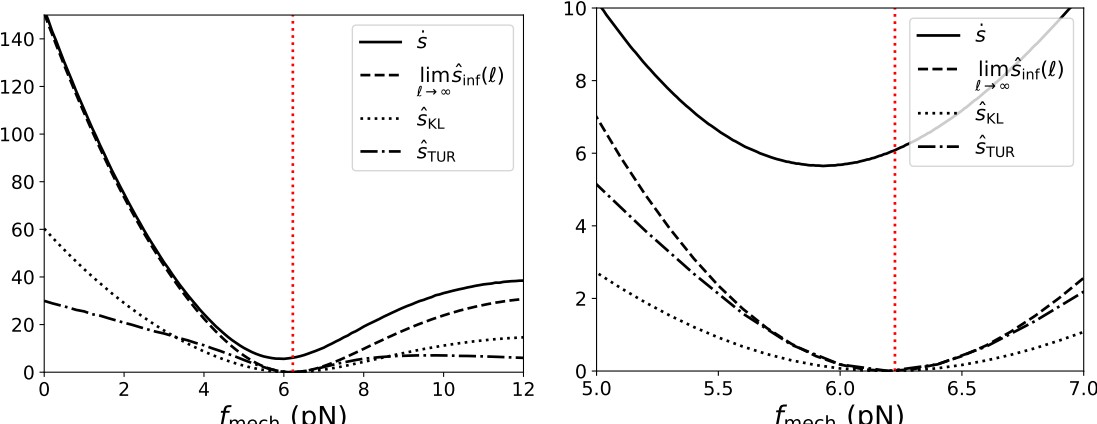

Figure 5: Three estimators of dissipation, viz., $\hat{s}_{\text{KL}}$, $\hat{s}_{\text{TUR}}$, and $\hat{s}_{\text{inf}} = \lim_{\ell\to\infty}\hat{s}_{\text{inf}}(\ell)$, are evaluated for the position $J_{\text{pos}}$ of a two-headed molecular motor. The estimators are plotted as a function of the mechanical force $f_{\text{mech}}$ together with the entropy production rate $\dot{s}$. The dynamics of the molecular motor is determined by the model in Sec. 7.1 and the parameters used are identical as in Fig. 4. The right figure is a closeup of the left figure around the stalled, nonequilibrium state, denoted by the vertical dotted line. All the rates are reported in $s^{-1}$ and $\hat{s}_{\text{TUR}}$ is evaluated at $t = 100s$ (other values of $t$ give similar results). The $\hat{s}_{\text{inf}} = \lim_{\ell\to\infty}\hat{s}_{\text{inf}}(\ell)$ is a plot of Eqs. (134)-(136).

where we omitted the explicit dependence of the rates on $f_{\text{mech}}$ in the first term. Notice that the average current $\bar{j}_{2\to5}$ is given by

$$\bar{j}_{2\to5} = p_{\text{ss}}(2)k_{2\to5} - p_{\text{ss}}(5)k_{5\to2}. \tag{136}$$

Figure 5 shows the quality of the three estimators $\hat{s}_{\text{inf}}$, $\hat{s}_{\text{KL}}$, and $\hat{s}_{\text{TUR}}$ when they are evaluated on the position $J = J_{\text{pos}}$ of the molecular motor. Remarkably, the estimator $\hat{s}_{\text{inf}} = \lim_{\ell\to\infty}\hat{s}_{\text{inf}}(\ell)$ based on the extreme value statistics of $J_{\text{pos}}$ captures a significant fraction of the dissipation, even in regimes far from thermal equilibrium where both the the thermodynamic uncertainty relation $\hat{s}_{\text{TUR}}$ and the Kullback-Leibler divergence $\hat{s}_{\text{KL}}$ capture a small proportion of the dissipation. Indeed, as discussed in Ref. [19], the thermodynamic uncertainty relation captures a negligible fraction of $\dot{s}$ in regimes far from thermal equilibrium, and $\hat{s}_{\text{KL}}$ is strongly biased when the statistics of the current are non-Markovian. However, as shown in Fig. 5, in contrast with $\hat{s}_{\text{TUR}}$ and $\hat{s}_{\text{KL}}$, the estimator $\hat{s}_{\text{inf}}$ accurately estimates entropy production rates far from thermal equilibrium. A notable exception is when the process is near a nonequilibrium stalled state (the vertical dotted line in Fig. 5), in which case none of the estimators capture the dissipation in the process.

As discussed, none of the estimators considered in this paper capture dissipation of the process near the stalling state. However, it should be emphasized that the estimator $\hat{s}_{\text{KL}}$ is a crude approximation of the Kullback-Leibler divergence Eq. (119), as it ignores nonMarkovian correlations in the trajectories of $J$. As shown in Refs. [51–53], by considering nonMarkovian effects, such as the statistics of the transition times along the edge, a fraction of the dissipation can be estimated, even at the stalling state.

# 9   Discussion

We have shown that the probability mass functions of infima of empirical, integrated, edge currents in nonequilibrium, stationary states of Markov jump processes are those of a geometric distribution. The geometric distribution is determined by two parameters, viz., the effective affinity $a^*_{x \to y}$, given by Eq. (68), and the probability $p_{\text{esc}}$ that the infimum equals zero, determined by Eqs. (89)-(90). In general, the latter probability does not admit a simple expression in terms of $a^*_{x \to y}$, except when the process starts in the source state $x$ of the observed transition.

The result Eq. (14) implies that the probability mass function of $J^{\text{inf}}_{x \to y}$ is that of a geometric distribution, independent of the underlying model. As we elaborate in Appendix H, this property is specific for edge currents, and hence can be used to test whether an observed current $J$ in a process $X$ — we assume here that the observer can measure $J$ but not $X$ — is an edge current. Similar tests of transition specificity have been proposed in Ref. [52, 53].

To derive the main results, we have identified the set of martingales $M_{x \to y}$ (see Eq. (69)) associated with the edge currents $J_{x \to y}$. The martingales $M_{x \to y}$ are Radon-Nikodym derivative processes, similar to other martingales studied in physics, such as the exponentiated negative entropy production [22–24, 46, 55–58] and the exponentiated housekeeping heat [30]. However, the conjugate probability measure defining the martingales $M_{x \to y}$ is not simply related to time-reversal (see Eq. (66)), as is the case for the entropy production. It will be interesting to find other examples of martingales in nonequilibrium physics, in particular, in physical contexts that we have not considered before. In this regard note the recent works [59, 60], which show that the mean equilibrium value of an unquenched spin in a fully connected spin model under progressive quenching is a martingale.

A marginal observer that only observes a current $J$ can estimate the average rate of entropy production $\dot{s}$ from the extreme value statistics of a current $J$ through the estimator $\hat{s}_{\text{inf}}(\ell)$ in the limit of large $\ell$ (see Eq. (3) for a definition of $\hat{s}_{\text{inf}}$); this estimator is smaller or equal than $\dot{s}$ and is equal to $\dot{s}$ when the observed current is proportional to the entropy production $S$ [10, 16, 19]. In this paper, we have shown that for the particular case when the observed current equals an edge current, i.e., $J = J_{x \to y}$, it holds that $\hat{s}_{\text{inf}} = a^*_{x \to y} \bar{j}_{x \to y}$, consistent with the thermodynamic interpretation of $a^*_{x \to y}$ as an effective affinity, see Refs. [32, 33, 35].

The estimator $\hat{s}_{\text{inf}}(\ell)$ lower bounds the rate of dissipation in the limit of large $\ell$. However, in this limit

$$p_-(\ell) = \exp\left(-a^* \ell [(1 + o(\ell)]\right) , \tag{137}$$

where the prefactor $a^* > 0$ is the effective affinity, and therefore the number of samples $n_s \sim 1/p_-$ required to estimate $p_-$ increases exponentially in $\ell$, see Ref. [19], which we have called the infinite threshold problem. In this Paper, we have shown that for edge currents the average rate of dissipation can be estimated from the extreme value statistics of a current at finite thresholds $\ell$ through the estimator $\hat{s}_{\text{inf}}$ (see Eq. (9)). This resolves, for the case of edge currents, the problem with infinite thresholds when estimating dissipation based on extreme value statistics, which is a special case of the first passage problem considered in Refs. [10, 16, 19]. This raises the interesting question whether the infinite threshold problem for estimators based on first passage processes can also be resolved for currents that are not edge currents.

# Acknowledgements

IN thanks C. Hyeon for a useful email communication and A. Raghu for carefully reading the manuscript. The research was supported by the National Research Fund Luxembourg (project

CORE ThermoComp C17/MS/11696700) and by the European Research Council, project NanoThermo (ERC-2015-CoG Agreement No. 681456).

# A  Radon-Nikodym derivative processes are martingales

We show that Radon-Nikodym derivative processes (as defined in Eq. (46) or(47)) are martingales. Since the conditions (i) and (ii) of the martingale definition in Sec. 4.4 are immediate, we focus on demonstrating the condition (iii), given by Eq. (59).

## A.1  Radon-Nikodym derivative processes as conditional expectations

Consider the filtered probability space $(\Omega, \mathscr{F}, \{\mathscr{F}_s\}_{s\in\mathbb{R}^+}, \mathbb{P})$ generated by the process $X$. Let $\mathbb{Q}$ be a second probability measure that is locally, absolutely continuous with respect to $\mathbb{P}$; note that we have dropped the $p^*$ and $q^*$ from $\mathbb{Q}$ and $\mathbb{P}$ as the arguments presented are general and not restricted to Markov jump processes.

We consider the Radon-Nikodym derivative process [28]

$$R(s) := \frac{\mathrm{d}\mathbb{Q}[X_0^s]}{\mathrm{d}\mathbb{P}[X_0^s]}, \tag{A.1}$$

with $s \in [0, t]$, and aim to show that

$$R(s) = \langle R(t)|X_0^s\rangle, \quad \forall s \in [0, t], \tag{A.2}$$

holds $\mathbb{P}$-almost surely, where $\langle \cdot|X_0^s\rangle$ is the conditional expectation with respect to the sub-$\sigma$-algebra $\mathscr{F}_s$ generated by the trajectory $X_0^s$.

To show that Eq. (A.2 ) holds, we first use the definition of the conditional expectation $\langle R(t)|X_0^s\rangle$, viz., $\langle R(t)|X_0^s\rangle$ is a random variable defined on $(\Omega, \mathscr{F}_s)$ [28] for which

$$\int_\Phi \langle R(t)|X_0^s\rangle\mathrm{d}\mathbb{P}[X_0^s] = \int_\Phi R(t)\mathrm{d}\mathbb{P}[X_0^t], \tag{A.3}$$

holds for all $\Phi \in \mathscr{F}_s$. Subsequently, we use that $R(t)$ is the Radon-Nikodym derivative (A.1 ) to write the right-hand side of Eq. (A.3 ) as

$$\int_\Phi R(t)\mathrm{d}\mathbb{P}[X_0^t] = \int_\Phi \mathrm{d}\mathbb{Q}[X_0^t]. \tag{A.4}$$

Marginalising the latter distribution leads to

$$\int_\Phi \mathrm{d}\mathbb{Q}[X_0^t] = \int_\Phi \mathrm{d}\mathbb{Q}[X_0^s], \tag{A.5}$$

and using that $R(s)$ is the Radon-Nikodym derivative (A.1 ), we obtain

$$\int_\Phi \mathrm{d}\mathbb{Q}[X_0^s] = \int_\Phi R(s)\mathrm{d}\mathbb{P}[X_0^s]. \tag{A.6}$$

Equations (A.3 )-(A.6 ) imply that Eq. (A.2 ) holds $\mathbb{P}$-almost surely, which we were meant to show.

## A.2  Martingale property of $R$

The martingale property (iii), given by Eq. (59), for $R(t)$ is a direct consequence of Eq. (A.2) and the tower property

$$R(s') = \langle R(t)|X_0^{s'}\rangle = \langle\langle R(t)|X_0^s\rangle|X_0^{s'}\rangle = \langle R(s)|X_0^{s'}\rangle, \quad \text{with} \quad 0 \le s' \le s \le t, \tag{A.7}$$

that holds for conditional expectations of a random variable [61].

## B  Derivation of Eq. (66)

We show that the martingale $M_{x \to y}$, given by Eq. (69), is the Radon-Nikodym derivative process Eq. (66), where $(X, \mathbb{R}_{\text{ss}})$ is the Markov jump process with rates $m_{x \to y}$ given by Eq. (67). To this aim, we use the formula (52), which is valid when the two conditions Eqs. (49) and (51) hold.

First, we verify (49), i.e., we verify that the exit rates

$$\sum_{v \in \mathcal{X};(v \ne u)} k_{u \to v} = \sum_{v \in \mathcal{X};(v \ne u)} m_{u \to v}. \tag{B.1}$$

Using the definition Eq. (67) for the rates $m_{u \to v}$ together with the fact that, by definition, $q_{\text{ss}}$ satisfies the stationary conditions

$$\sum_{v \in \mathcal{X};(v \ne u)} q_{\text{ss}}(v)\ell_{v \to u} = q_{\text{ss}}(u) \sum_{v \in \mathcal{X};(v \ne u)} \ell_{u \to v}, \tag{B.2}$$

for the Markov jump process $(X, \mathbb{Q}_{\text{ss}})$ with rates $\ell_{u \to v}$ given by Eq. (61), we recover (B.1).

Second, we verify (51), which follows readily from the definition of the rates (67).

Hence we can use Eq. (52), for the Radon-Nikodym derivative of $\mathbb{R}_{\text{ss}}$ with respect to $\mathbb{P}_{\text{ss}}$, to obtain

$$\frac{d\mathbb{R}_{\text{ss}}[X_0^t]}{d\mathbb{P}_{\text{ss}}[X_0^t]} = \frac{q_{\text{ss}}(X(0))}{p_{\text{ss}}(X(0))} \exp\left(\frac{1}{2} \sum_{(u,v) \in \mathcal{E}} J_{u \to v}(\omega, t) \ln \frac{m_{u \to v}}{k_{u \to v}}\right). \tag{B.3}$$

Consequently, using the definition (67) in (B.3), we obtain

$$\frac{d\mathbb{R}_{\text{ss}}[X_0^t]}{d\mathbb{P}_{\text{ss}}[X_0^t]} = \frac{q_{\text{ss}}(X(0))}{p_{\text{ss}}(X(0))} \exp\left(\frac{1}{2} \sum_{(u,v) \in \mathcal{E}} J_{u \to v}(\omega, t) \ln \frac{q_{\text{ss}}(v)p_{\text{ss}}^{x,y}(u)}{q_{\text{ss}}(u)p_{\text{ss}}^{x,y}(v)} + J_{x \to y} \ln \frac{p_{\text{ss}}^{x,y}(y)k_{y \to x}}{p_{\text{ss}}^{x,y}(x)k_{x \to y}}\right), \tag{B.4}$$

where the $J_{x \to y} \ln p_{\text{ss}}^{x,y}(x)/p_{\text{ss}}^{x,y}(y)$ in the first term of the exponent cancels out with the $J_{x \to y} \ln p_{\text{ss}}^{x,y}(y)/p_{\text{ss}}^{x,y}(x)$ in the second term of the exponent. In addition, identifying

$$q_{\text{ss}}(X(0)) \exp\left(\frac{1}{2} \sum_{(u,v) \in \mathcal{E}} J_{u \to v}(\omega, t) \ln \frac{q_{\text{ss}}(v)}{q_{\text{ss}}(u)}\right) = q_{\text{ss}}(X(t)), \tag{B.5}$$

and

$$p_{\text{ss}}^{x,y}(X(t)) \exp\left(\frac{1}{2} \sum_{(u,v) \in \mathcal{E}} J_{u \to v}(\omega, t) \ln \frac{p_{\text{ss}}^{x,y}(u)}{p_{\text{ss}}^{x,y}(v)}\right) = p_{\text{ss}}^{x,y}(X(0)), \tag{B.6}$$

in (B.4), we obtain

$$\frac{d\mathbb{R}_{\text{ss}}[X_0^t]}{d\mathbb{P}_{\text{ss}}[X_0^t]} = \frac{p_{\text{ss}}^{x,y}(X(0))q_{\text{ss}}(X(t))}{p_{\text{ss}}^{x,y}(X(t))p_{\text{ss}}(X(0))} e^{-a_{x \to y}^* J_{x \to y}(t)} = M_{x \to y}(t), \tag{B.7}$$

and thus according to (69) we find (66), which is our desired result.

## C  Microscopic and effective affinities in unicyclic systems

We compare the microscopic affinities $a_{x \to y}$, as defined by Eq. (40), with the effective affinities $a^*_{x \to y}$, as defined by Eq. (68), in unicyclic systems. In particular, we consider systems described by the following Master equation,

$$\partial_t p(u;t) = p(u+1;t)k_{u+1 \to u} + p(u-1;t)k_{u-1 \to u} - (k_{u \to u+1} + k_{u \to u-1})p(u;t), \qquad \text{(C.1)}$$

where $u \in \mathcal{X} = \{1, 2, \dots, \ell\}$, and in Eq. (C.1) it should be understood that $0 = \ell$ and $\ell + 1 = 1$.

At stationarity, $\partial_t p(u;t) = 0$, such that the edge currents given by Eq. (39) obey

$$\bar{j}_{u-1 \to u} = \bar{j}_{u \to u+1} = \bar{j}, \qquad \text{(C.2)}$$

for all $u \in \mathcal{X}$. Using Eq. (C.2) in Eq. (58), we obtain for the rate of dissipation,

$$\dot{s} = a\bar{j}, \qquad \text{(C.3)}$$

where $a$ is identified as the "true", macroscopic affinity

$$a = \sum_{u=1}^{\ell} a_{u \to u+1}. \qquad \text{(C.4)}$$

Note that the macroscopic affinity $a$ is the sum of all microscopic affinities $a_{u \to v}$. Hence, in general, the microscopic affinities contribute a small part of the total affinity. For example, when $k_{u \to u+1} = k_+$ for all $u \in \mathcal{X}$, and $k_{u \to u-1} = k_-$ for all $u \in \mathcal{X}$, then

$$a_{u \to u+1} = \ln \frac{k_+}{k_-}, \qquad \text{(C.5)}$$

and

$$a = \ell \ln \frac{k_+}{k_-}. \qquad \text{(C.6)}$$

Let us now determine the effective affinities $a^*_{x \to y}$ of unicyclic Markov processes described by Eq. (C.1). To determine $a^*_{x \to y}$, we need to determine the values $p^{x,x+1}_{\mathrm{ss}}(x)$ and $p^{x,x+1}_{\mathrm{ss}}(x+1)$ of the stationary distributions $p^{x,x+1}_{\mathrm{ss}}(u)$ solving

$$0 = p^{x,x+1}_{\mathrm{ss}}(u+1)k_{u+1 \to u} + p^{x,x+1}_{\mathrm{ss}}(u-1)k_{u-1 \to u} - (k_{u \to u+1} + k_{u \to u-1})p^{x,x+1}_{\mathrm{ss}}(u), \qquad \text{(C.7)}$$

for all $u \in \mathcal{X} \setminus \{x, x+1\}$,

$$k_{x \to x-1} p^{x,x+1}_{\mathrm{ss}}(x) = p^{x,x+1}_{\mathrm{ss}}(x-1)k_{x-1 \to x}, \qquad \text{(C.8)}$$

and

$$k_{x+1 \to x+2} p^{x,x+1}_{\mathrm{ss}}(x+1) = p^{x,x+1}_{\mathrm{ss}}(x+2)k_{x+2 \to x+1}. \qquad \text{(C.9)}$$

Solving the Eqs. (C.7 -C.9), we obtain

$$p^{x,x+1}_{\mathrm{ss}}(x) = p_0 \prod_{u=1; u \neq x}^{\ell} \frac{k_{u \to u+1}}{k_{u+1 \to u}}, \qquad \text{(C.10)}$$

and

$$p^{x,x+1}_{\mathrm{ss}}(x+1) = p_0, \qquad \text{(C.11)}$$

where $p_0$ is a normalisation constant. Substitution of Eqs. (C.10) and (C.11) in the definition (68) of $a^*_{x \to x+1}$ yields

$$a^*_{x \to x+1} = \sum_{u=1}^{\ell} a_{u \to u+1} = a. \qquad \text{(C.12)}$$

Hence, the effective affinity in a unicyclic system equals the macroscopic affinity.

# D  Derivation of $a^*_{x\to y}\langle J_{x\to y}(t)\rangle_{\mathrm{ss}} \geq 0$

We use the $\mathbb{P}_{\mathrm{ss}}$-martingale $M_{x\to y}(t)$, given by Eq. (69) to show that $a^*_{x\to y}\langle J_{x\to y}(t)\rangle_{\mathrm{ss}} \geq 0$.
    Indeed, since $M_{x\to y}(t)$ is a $\mathbb{P}_{\mathrm{ss}}$-martingale, it holds that

$$\langle M_{x\to y}(t)\rangle_{\mathrm{ss}} = \langle M_{x\to y}(0)\rangle_{\mathrm{ss}} = 1. \tag{D.1}$$

In addition, since

$$M_{x\to y}(t) = e^{-a^* J_{x\to y}(t) + O_t(1)}, \tag{D.2}$$

where the big-O notation $O_t(1)$ denotes an arbitrary function of $t$ that is bounded, it holds that

$$\langle M_{x\to y}(t)\rangle_{\mathrm{ss}} = \langle e^{-a^*_{x\to y} J_{x\to y}(t) + O_t(1)}\rangle_{\mathrm{ss}} = 1. \tag{D.3}$$

Applying Jensen's inequality

$$\langle e^{-a^*_{x\to y} J_{x\to y}(t) + O_t(1)}\rangle_{\mathrm{ss}} \geq e^{-a^*_{x\to y}\langle J_{x\to y}(t)\rangle_{\mathrm{ss}} + O_t(1)}, \tag{D.4}$$

and using

$$\langle J_{x\to y}(t)\rangle_{\mathrm{ss}} = \bar{j}_{x\to y}\, t, \tag{D.5}$$

with $\bar{j}_{x\to y} \in \mathbb{R}$ the average current, we obtain

$$a^*_{x\to y}\bar{j}_{x\to y}\, t + O_t(1) \geq 0. \tag{D.6}$$

Since $O_t(1)/t \to 0$, it holds that

$$a^*_{x\to y}\langle J_{x\to y}(t)\rangle_{\mathrm{ss}} \geq 0. \tag{D.7}$$

According to Eq. (D.7), $\langle J_{x\to y}(t)\rangle_{\mathrm{ss}}$ changes sign when $a^*_{x\to y} = 0$. Hence, $a^*_{x\to y} = 0$ if and only if $\langle J_{x\to y}(t)\rangle_{\mathrm{ss}} = 0$.

# E  Equilibrium distribution in the six state model

The equilibrium state of the six-state model defined in Sec. 7 is

$$p_{\mathrm{eq}}(1) = \frac{1}{\mathcal{N}}, \quad p_{\mathrm{eq}}(2) = \frac{1}{\mathcal{N}}\sqrt{\frac{k_{5\to4}(0)}{k_{2\to1}(0)}\frac{k_{4\to3}(0)}{k_{3\to4}(0)}\frac{k_{3\to2}(0)}{k_{2\to3}(0)}}, \quad p_{\mathrm{eq}}(3) = \frac{1}{\mathcal{N}}\sqrt{\frac{k_{5\to4}(0)}{k_{2\to1}(0)}\frac{k_{4\to3}(0)}{k_{3\to4}(0)}},$$

$$p_{\mathrm{eq}}(4) = \frac{1}{\mathcal{N}}\sqrt{\frac{k_{5\to4}(0)}{k_{2\to1}(0)}}, \quad p_{\mathrm{eq}}(5) = \frac{1}{\mathcal{N}}\frac{k_{4\to3}(0)}{k_{3\to4}(0)}\frac{k_{3\to2}(0)}{k_{2\to3}(0)}, \quad p_{\mathrm{eq}}(6) = \frac{1}{\mathcal{N}}\frac{k_{4\to3}(0)}{k_{3\to4}(0)}, \tag{E.1}$$

where the normalisation constant is

$$\mathcal{N} = 1 + \sqrt{\frac{k_{5\to4}(0)}{k_{2\to1}(0)}\frac{k_{4\to3}(0)}{k_{3\to4}(0)}\frac{k_{3\to2}(0)}{k_{2\to3}(0)}} + \sqrt{\frac{k_{5\to4}(0)}{k_{2\to1}(0)}\frac{k_{4\to3}(0)}{k_{3\to4}(0)}}$$

$$+ \sqrt{\frac{k_{5\to4}(0)}{k_{2\to1}(0)}} + \frac{k_{4\to3}(0)}{k_{3\to4}(0)}\frac{k_{3\to2}(0)}{k_{2\to3}(0)} + \frac{k_{4\to3}(0)}{k_{3\to4}(0)}. \tag{E.2}$$

## F  Parameters for the six-state model of Kinesin-1

We specify the parameters that we use in Sec. 7.2 for the six-state model for two-headed molecular motors, as visualised in Fig. 2.

We use the parametrisation of the rates given by Eqs. (101), (102), (103), (105), and (106). We set the parameters $k_{i\to j}(0)$, $\theta$, and $\chi_{ij}$ to the same values as those given in Refs. [37], which were obtained by fitting single molecule motility data of Kinesin-1 to the six-state model, except for the parameter $k_{5\to2}(0)$, which we set to (107) such that the model obeys detailed balance when $f_{\text{mech}}$ and [ATP] are given by Eq. (108). Note that from the pragmatic point of view of modelling the dynamics of Kinesin-1, this does not make much of a difference, as in our case $k_{5\to2}(0) \approx 1.13\,\text{s}^{-1}$ while in Ref. [37] $k_{5\to2}(0) = 1.1\,\text{s}^{-1}$. However, from a theoretical point of view, it is desirable to have an equilibrium point in the model as it allows us to study the properties of the system near equilibrium.

Concretely, we set $\theta = 0.61$, $\chi_{12} = 0.15$, $\chi_{56} = 0.0015$, $\chi_{61} = 0.11$, $k_{2\to1}(0) = 4200\,s^{-1}$, $k_{2\to5}(0) = 1.6 \times 10^6\,s^{-1}$, $k_{5\to2}(0) = 1.1\,s^{-1}$, $k_{5\to6}(0) = 190\,s^{-1}$, $k_{6\to5}(0) = 10\,s^{-1}$, $k_{6\to1}(0) = 250\,s^{-1}$, $k_{1\to6}(0) = 230\,s^{-1}$, $k_{5\to4}(0) = 2.1 \times 10^{-9}\,s^{-1}$, and we set $k_{1\to2}(0) = k_{1\to2}^{\text{bi}}(0)[\text{ATP}]$ with $k_{1\to2}^{\text{bi}}(0) = 2.8\,\mu M^{-1}s^{-1}$.

Note that we have set $k_{3\to2}(0) = k_{6\to5}(0)$, $k_{2\to3}(0) = k_{5\to6}(0)$ $k_{3\to4}(0) = k_{6\to1}(0)$, $k_{4\to3}(0) = k_{1\to6}(0)$, $k_{4\to5}(0) = k_{1\to2}(0)$, $\chi_{23} = \chi_{56}$, $\chi_{34} = \chi_{61}$, and $\chi_{45} = \chi_{12}$.

The molecular motor step size is set to the length of a tubulin dimer (the subunit that forms the microtubule filament that is the substrate to which kinesin binds), viz., $\delta = 8\,\text{nm}$, and the environment is set to room temperature, i.e., $T_{\text{env}} = 298 \times 1.38 \times 10^{-23}$ J.

## G  The stationary distribution of the six-state model in the absence of the $2 \to 5$ link

The stationary distribution $p_{\text{ss}}^{2,5}(x)$ of the six-state model in the absence of the $2 \to 5$ and $5 \to 2$ transitions, as defined in Sec. 7.1, is given by:

$$
\begin{aligned}
p_{\text{ss}}^{2,5}(1) &= \frac{1}{\mathcal{N}^{2,5}} k_{1\to2} k_{2\to3} k_{3\to4} \left( k_{2\to3} k_{3\to4} + k_{2\to1} \left[ k_{3\to2} + k_{3\to4} \right] \right) \\
&\quad + \frac{1}{\mathcal{N}^{2,5}} k_{2\to1} k_{3\to2} k_{4\to3} \left( k_{2\to3} k_{3\to4} + \left[ k_{3\to2} + k_{3\to4} \right] k_{5\to4} \right), \\
p_{\text{ss}}^{2,5}(2) &= \frac{1}{\mathcal{N}^{2,5}} \left( \left[ k_{1\to2} \left\{ k_{3\to2} + k_{3\to4} \right\} + k_{3\to2} k_{4\to3} \right] \left[ k_{1\to2} k_{2\to3} k_{3\to4} + k_{3\to2} k_{4\to3} k_{5\to4} \right] \right), \\
p_{\text{ss}}^{2,5}(3) &= \frac{1}{\mathcal{N}^{2,5}} k_{1\to2} k_{2\to3}^2 k_{3\to4} \left( k_{1\to2} + k_{4\to3} \right) \\
&\quad + \frac{1}{\mathcal{N}^{2,5}} k_{4\to3} k_{5\to4} \left( k_{1\to2} k_{2\to3} \left[ k_{3\to2} + k_{3\to4} \right] + \left[ k_{2\to1} + k_{2\to3} \right] k_{3\to2} k_{4\to3} \right), \\
p_{\text{ss}}^{2,5}(4) &= \frac{1}{\mathcal{N}^{2,5}} k_{3\to2} k_{4\to3} k_{5\to4} \left( k_{2\to3} k_{3\to4} + k_{2\to1} \left[ k_{3\to2} + k_{3\to4} \right] \right) \\
&\quad + \frac{1}{\mathcal{N}^{2,5}} k_{1\to2} k_{2\to3} k_{3\to4} \left( k_{2\to3} k_{3\to4} + k_{5\to4} \left[ k_{3\to2} + k_{3\to4} \right] \right), \\
p_{\text{ss}}^{2,5}(5) &= \frac{1}{\mathcal{N}^{2,5}} \left( \left[ k_{1\to2} \left\{ k_{3\to2} + k_{3\to4} \right\} + k_{3\to2} k_{4\to3} \right] \left[ k_{1\to2} k_{2\to3} k_{3\to4} + k_{2\to1} k_{3\to2} k_{4\to3} \right] \right), \\
p_{\text{ss}}^{2,5}(6) &= \frac{1}{\mathcal{N}^{2,5}} \left( k_{1\to2}^2 k_{2\to3}^2 k_{3\to4} + k_{2\to1} k_{3\to2} k_{4\to3}^2 \left[ k_{2\to3} + k_{5\to4} \right] \right) \\
&\quad + \frac{1}{\mathcal{N}^{2,5}} k_{1\to2} k_{2\to3} k_{4\to3} \left( k_{2\to3} k_{3\to4} + k_{2\to1} \left[ k_{3\to2} + k_{3\to4} \right] \right),
\end{aligned}
$$

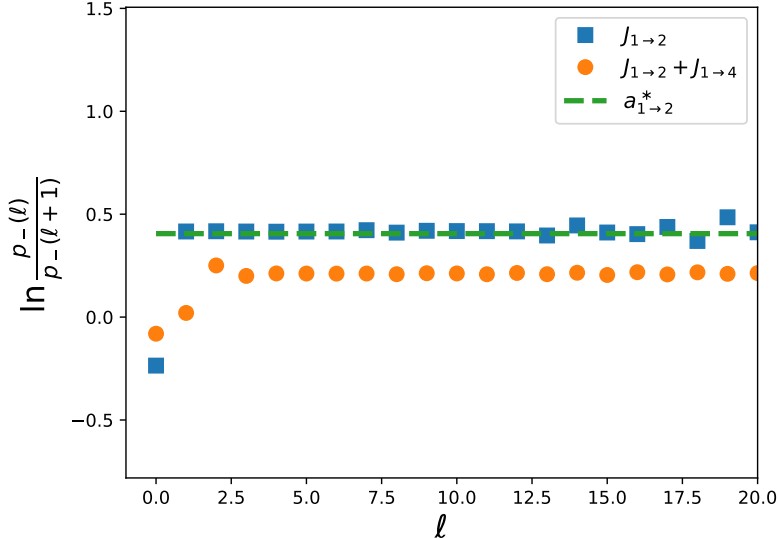

Figure 6: The modified infimum ratio $\hat{s}_{\text{inf}}$ as a function of $\ell$ for the edge current $J_{1\to 2}$ (blue squares) and the sum $J_{1\to 2} + J_{1\to 4}$ of two edge currents (orange circles) in the stationary Markov jump process with four states and transition rates given by Eq. (H.3) with parameters $c = 1.5$ and $b = 1/e$. Markers are simulation results for $\hat{s}_{\text{inf}}$ obtained from $1e+7$ simulation runs over a time interval $t \in [0, 1e+4]$, which are compared with the effective affinity $a_{1\to 2}^*$ given by Eq. (H.5).

where

$$
\begin{aligned}
\mathcal{N}^{2,5} \;=\; & 2k_{1\to 2}^2 k_{2\to 3} k_{3\to 4}\left(k_{2\to 3} + k_{3\to 2} + k_{3\to 4}\right) \\
& + k_{1\to 2}\left(k_{3\to 2}k_{4\to 3} + k_{2\to 3}[k_{3\to 4} + k_{4\to 3}]\right) \\
& \quad \times \left(2k_{2\to 3}k_{3\to 4} + k_{2\to 1}[k_{3\to 2} + k_{3\to 4}] + [k_{3\to 2} + k_{3\to 4}]k_{5\to 4}\right) \\
& + k_{3\to 2}k_{4\to 3}\left(k_{2\to 1}k_{2\to 3}k_{3\to 4} + k_{2\to 1}[k_{2\to 3} + k_{3\to 2}]k_{4\to 3}\right. \\
& \left. + k_{5\to 4}\left[k_{2\to 3}k_{3\to 4} + \{k_{2\to 3} + k_{3\to 2}\}k_{4\to 3} + 2k_{2\to 1}\left\{k_{3\to 2} + k_{3\to 4} + k_{4\to 3}\right\}\right]\right), \quad \text{(G.1)}
\end{aligned}
$$

is the normalisation constant.

# H The modified infimum ratio for empirical, integrated currents that are not edge currents

For empirical, integrated, edge currents, $J_{x\to y}$, it holds that for all $\ell \in \mathbb{N}$ the modified infimum ratio

$$
\hat{s}_{\text{inf}} = \ln \frac{p_-(\ell)}{p_-(\ell + 1)} = a_{x\to y}^* \tag{H.1}
$$

is a constant independent of $\ell$; notice that for $\ell = 0$ the modified infimum ratio takes a value different from $a_{x\to y}^*$. Recall that in Eq. (H.1)

$$
p_-(\ell) = \mathbb{P}_{\text{ss}}\left(J_{x\to y}^{\text{inf}} \le -\ell\right). \tag{H.2}
$$

In the asymptotic limit of large $\ell$, i.e., $\ell \gg 1$, the constancy of $\hat{s}_{\text{inf}}$ holds for generic currents $J$. However, at finite $\ell$, the modified infimum ratio is, in general, not a constant. Consequently, the constancy of $\hat{s}_{\text{inf}}$ for $\ell \in \mathbb{N}$ can be used to identify whether a current $J$ is an edge current.

Let us illustrate this on a simple model. Consider a four state Markov jump process for which $X(t) \in \mathcal{X} = \{1, 2, 3, 4\}$ and with

$$k_{u \to v} = \begin{cases} 1, & \text{if } u > v, \\ c b^{v-u}, & \text{if } u < v, \end{cases} \tag{H.3}$$

where $c, b > 0$, and $u, v \in \{1, 2, 3, 4\}$.

The stationary distribution of this model is given by

$$
\begin{aligned}
p_{\text{ss}}(1) &= \frac{1}{\mathcal{N}} \frac{2 + bc + b^2 c}{b^3 c (1 + c + bc + b^2 c)}, \quad p_{\text{ss}}(2) = \frac{1}{\mathcal{N}} \frac{2 + b + b^2}{b^2 (1 + c + bc + b^2 c)}, \\
p_{\text{ss}}(3) &= \frac{1}{\mathcal{N}} \frac{1}{b}, \quad p_{\text{ss}}(4) = \frac{1}{\mathcal{N}},
\end{aligned}
\tag{H.4}
$$

where $\mathcal{N}$ is a normalisation constant.

In Fig 6, we present numerical simulation results for the modified infimum ratio $\hat{s}_{\text{inf}}$ of the edge current $J_{1 \to 2}$ with effective affinity

$$a_{1 \to 2}^* = \ln c, \tag{H.5}$$

and we also present $\hat{s}_{\text{inf}}$ evaluated on the infima of the current $J_{1 \to 2} + J_{1 \to 4}$. Figure 6 shows that the modified infimum ratio of the edge current is constant and equal to the effective affinity, while the modified infimum ratio of $J_{1 \to 2} + J_{1 \to 4}$ is nonconstant for small values of $\ell$ before saturating to a constant value at intermediate values of $\ell$.

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
