# Peer review of "Extreme value statistics of edge currents in Markov jump processes and their use for entropy production estimation"

_SciPost Physics, doi:SciPost Phys. 14, 131 (2023)_

## Round 2 · Referee Report · Anonymous (Referee 1) · 2022-9-22

Strengths

1) The paper provides a timely, relevant study of martingale methods applied to the study of currents along a single visible observed edge (as introduced in Ref. [30]).

2) The first two chapters provide a clear, intuitive introduction into the methodology and physical significance of this work.

Weaknesses

1) Chapter 8 compares the method to recent entropy estimators, which are quoted but not discussed properly. Thus, the conclusions might give a misleading picture of recent entropy estimation methods.

2) The physical significance of the results beyond the summary might be difficult to grasp due to the heavy mathematical formalism introduced throughout the work.

Report

The study of partially accessible Markov networks is of major interest
in stochastic thermodynamics. Gaining new insights into statistics
of observables and estimators for entropy production is not only a
timely challenge but also crucial from an operational point of view.

The present work adapts a transition-based description for partially
accessible Markov networks to apply known results from martingale theory
and extant entropy estimators to currents counting transitions along a
single edge in a Markov network. The idea is not novel, but generalizes
previous results that are based on martingale theory (e.g. Ref. [22]).

In a numerical case study for a molecular motor model, the corresponding
entropy estimator is compared to known estimators like the thermodynamic
uncertainty relation, the first passage ratio and a naive bound based
on the Kullback-Leibler divergence. The deduced entropy estimator is
interpreted as an effective affinity, a concept already introduced
in Ref. [30].

The manuscript is well written, both in terms of grammar and
mathematical rigor. It provides an introduction and a summary of the
main results that give the reader a clear overview over the paper and
how it relates to recent literature. While the theoretical sections of
the paper are kept at a quite technical level, the paper gives explicit
demonstrations of the main results in a numerical case study.

Requested changes

Major concerns:

  1. It is unclear to me how the paper concludes that the derived estimator "can be significantly more accurate than those based on ... Kullback-Leibler divergences".

Equation (112) is the only one in the manuscript that is claimed to be an estimator based on the Kullback-Leibler divergence. However, it is certainly not the only estimator of this sort, since more sophisticated versions have been proved e.g. in Ref. [49] and particularly in Ref. [48], which discusses an entropy estimator for the identical molecular motor model.

In light of these recent advances, it is, in my opinion, not justified to claim greater accuracy without a comparison to these recent bounds. The authors should either clarify these misleading conclusions or include these more recent entropy estimators in the numerical study. (see also the minor remark below)

Minor remarks:

  1. Two novel entropy estimators, the infimum ratio and the modified infimum ratio, are introduced. From an operational point of view, it is interesting to know how much statistics is actually needed for these estimators, i.e. how long should the trajectory be for comparable error/bias? Is there a connection to the results obtained in [18]?

  2. The math in Ch. 5 seems to be very similar to the derivations leading to the "informed partial" estimator in Gili Bisker et al J. Stat. Mech. (2017) 093210. Both incorporate the effective affinity of Ref [30] and the stalling distribution (cf. the paragraph following eq. (63)). How deep is this connection? More concrete, it might be more appropriate to numerically compare to the entropy estimators discussed in Bisker et al. These estimators seem more closely related to the present work than those that are based on waiting times.

  • validity: high
  • significance: high
  • originality: high
  • clarity: good
  • formatting: excellent
  • grammar: excellent

Author:  Izaak Neri  on 2023-01-10  [id 3226]

(in reply to Report 1 on 2022-09-22)
Category:
question

The attachment contains our Reply to the Referee's comments .

Attachment:

RefereeReply_H8BNyfb.pdf

---

## Round 2 · Referee Report · Anonymous (Referee 2) · 2022-10-30

Strengths

See report

Weaknesses

See report

Report

The authors consider stochastic processes in Markov networks and, in particular, the statistics of the fluctuating, time-integrated current across an individual edge of a network. For a statistical characterisation of these currents, this work focuses on the distribution of the infimum of the current, against an overall positive trend. Using methods from the theory of martingales, it is shown that this distribution is always a geometric one. The strength of this work is that it provides an operationally accessible way to evaluate the effective affinity of a single edge of a network, of which all other edges are invisible. This unlocks the possibility to apply bounds on the overall rate of entropy production that have been derived before (by one of the present authors, in Refs. [15,18]) and where this effective affinity enters as well. Unlike in this previous work, it is now possible to evaluate the effective affinity from only typical fluctuations, circumventing the unpractical procedure involving the limit of exponentially unlikely fluctuations. The distribution of the infimum and the resulting bound on the entropy production (compared to established bounds) are illustrated numerically for a well-known model for a molecular motor. Remarkably, this bound on the entropy production can be much tighter than the established bounds.

The article is well written, with the right attention to detail, and the results are indeed relevant. I appreciate the splitting of the work into two "companion papers", along with Ref. [1]. This manuscript is self-contained (and I understand the other one is so as well), and the equivalence of the results is discussed in Sec. 2.2. Attempting to sketch both derivations (which are complementary) in one manuscript would have overwhelmed the reader.

A few points remained unclear to me and there are a number of minor points that need to be fixed. I expect that the authors will be able to address these points and change the manuscript adequately, such that the paper will be ready for acceptance.

a.

In the introduction, I'd find it important to mention what class of physical systems is being considered, to provide some context for the first few paragraphs. The bound mentioned in the first paragraph is fairly universal, yet systems with inertia or coherent quantum dynamics can break it.

b.

It would be good to discuss in more detail the relation between the "effective affinity" and the "true affinity" (whatever that is). When are the two the same? I suspect that a*(x->y) is more related to the overall cycle affinity (in case of a unicyclic system) than to the true edge-affinity a(x->y), even though the notation would suggest some relation to the latter.

c.

p.7: "the system attains the stalling state, which according to the marginal observer is indistinguishable
from equilibrium [1]". It may be distinguishable if the marginal observer does the right analysis, e.g., detect coherent oscillations in the autocorrelation function of the current. Also, I couldn't find the term "marginal observer" in [1], so I couldn't find what it has to say about this.

d.

In Sec. 3, the concept of a current being "proportional" to the entropy needs to be specified better. Is it exactly proportional or up to finite boundary terms?

e.

I don't see how to get from (44) to (46); how can the current and the factor 1/2 be identified? It seems to me that we need to have l(u->v)=k(v->u) (which is the case in the following applications of this formula).

f.

Sec. 6.4 (and also mentioned elsewhere in this manuscript): I doubt that it is possible to have "all microscopic affinities zero except one". When the affinity ln((p_u k(u->v))/(p_v k(v->u)) is zero, then also the current (p_u k(u->v))-(p_v k(v->u)) is zero. But, because of the Kirchhoff rules for stationary currents, it is not possible to have a current through one edge, without a backflow through some other edge(s). In Sec. 7.2.1 it looks like the affinity is only the log-ratio of transition rates (without the contribution from the stationary distribution). In that case, it may be possible to have only one "affinity" equal to zero. But I'm not sure whether the argument from 6.4 then still applies.

g.

In Sec. 7, I'd welcome some more discussion. Beyond confirming the geometric distribution, what other information can be extracted from this kind of measurement? For example, is it possible to decide whether the observed current stems from a single or from several edges? In the latter case, is there a strong deviation from the geometric distribution?

h.

It is not clear to me how the bound derived from the infimum statistics can be so much stronger than the TUR. In which cases can the bound be saturated? (without the TUR saturating at the same time?) In Ref. [15], it is stated (below Eq. (19) there) that the equivalent bound on the splitting probability follows from the TUR. So how can it be stronger?
In the caption of Fig. 5, it is stated that the TUR is evaluated at 100s, which I assume is long enough to capture the long-time limit. Yet, the finite-time TUR can (in some cases) be stronger. How would the TUR evaluated at the optimal time interval compete with the bound from the infimum statistics? I think this would be a "fair game", because also the latter takes into account fluctuations on finite time scales.

Minor points:

1.

abstract: "current" -> "integrated current" (the more familiar concept of a current is that of a rate, where the extreme value would be ill-defined)

2.

abstract: "estimate" -> "bound from below" (If I'm not mistaken, the bound can still be way off. For an "estimate" I would expect a guess with a lower and upper confidence interval)

3.

"largest excursion against its average flow" sounds like max(|J-<J>|), which would not make sense. I think this can be made more precise with a better word choice (or dropped altogether, the explanation before and the Figure already make the concept clear).

4.

In which sense are (1) and (2) "analogous"? Can (1) be derived from (2)?

5.

"Interestingly, far from equilibrium, s_inf captures ..." What is the meaning of s_inf without the argument l? Is the limit to infinity implied, or is l arbitrary?

6.

The notation s_inf(l) in (9), (10) is somewhat confusing, since this quantity does not depend on l.

7.

Below (16): I'm confused about the difference between "geometric" and "exponential" distribution. According to wikipedia, they (only?) differ by whether the support is discrete or continuous.

8.

Eq. (19): The $\to\infty$ is typeset as subscript (please check this throughout, I've also seen this elsewhere, e.g. (125), (126)).

9.

is the f- in Eq. (23) just 1-p^esc? If so, say so.

Sec. 3: "Such an observer thinks that the observed current J is proportional to entropy production." Just because (s)he is naive, or is that simplest assumption indeed consistent with all observations?

10.

Eq. (31), typo a->c ?

11.

Typo: Full stop at end of sentence including (33).

12.

Beginning of Sec. 4.1: A few examples for "events" may be helpful ("event" sounds rather time-local, but I believe sigma can depend on several times)

13.

typo: "devined"

14.

Around (37), (38): Difference between Markov process and Markov chain?

15.

I never understood why it's called Radon-Nikodym "derivative" (and not "ratio"). For a derivative I'd expect the notion of a small difference. What is the meaning of "dQ" and "dP" in (44)?

16.

Below (52): "than" -> "then"

17.

Before (54), "|M(t)|<c". Does c need to be independent of T?

18.

"which is nota bene different from p_ss^xy": worth pointing out that it is also different from p_ss

19.

Before (58): Typo P_q_ss -> P_p_ss ?

20.

In (60): is R_q_ss=Q_ss\circ Theta?

21.

"Lastly, introducing the effective microscopic affinity": is this the same quantity as in Ref. [30]?

22.

In what sense is the meaning of the effective affinity "kinematic" (as opposed to the true efficiency?)

23.

p. 15: Notation switches from J_{x->y}(T) to J(T). Make consistent or at least introduce the abbreviation.

24.

I think "generic" should be replaced by "general" in Sec. 6.2. Surely, the initial condition x, which is not "generic", is included in this general case.

25.

Between (81) and (82), the sentence starting with "Therefore ..." appears out of context with the preceding sentence ("Notice ...").

26.

Below (85): I believe this should read "for a *near* stalled current", and a* should be approximately 0, otherwise p_esc and j would be zero exactly.

27.

Before (95): It is not clear how the parameter dependence "does not contribute to dissipation". The affinity doesn't depend on the parameter, but the kinetics depends on it, hence the dissipation rate (if this can be equated to "dissipation") indirectly depends on the parameter as well.

28.

Caption Fig. 3: "lines"->"line" (I just see one line)

29.

Sec. 7.2.3: typo "stalled stated"

30.

p. 24: "However, when the statistics of the current J contain strong non-Markovian effects and when J is not proportional to the entropy production S, than s_KL provides a poor estimate ..." With strong non-Markovian effects, can s_KL even be measured? (Before, it is stated that non-Markovian effects are ignored).

31.

I am not sure whether I understand the distinction between s hat and s double-hat. If (121) "applies" only to a single edge, should't it be a double-hat? I understand that in any case, the estimated entropy production is that of the overall network, not just a single edge.
  • validity: high
  • significance: high
  • originality: high
  • clarity: high
  • formatting: perfect
  • grammar: perfect

Author:  Izaak Neri  on 2023-01-10  [id 3225]

(in reply to Report 2 on 2022-10-30)
Category:
question

The attachment contains a reply to the Referee's comments.

Attachment:

RefereeReply.pdf

---

## Round 3 · Referee Report · Anonymous (Referee 1) · 2023-1-26

Strengths

see report

Weaknesses

see report

Report

The adjustments in this revised manuscript clarify most of my previously
raised concerns. Taking additionally into account the changes made based
on the report of referee 2, the present version of the manuscript
shows substantial improvements over the previous one. This work is now,
in my opinion, a valuable addition to the field highlighting the
operational significance of martingale theory for inference problems in
partially accessible Markov jump processes.

As a last remark, I have to iterate that the proposed entropy
estimator, dubbed the modified infimum ratio, is in fact the
"informed partial" estimator from Gili Bisker et al J. Stat. Mech.
(2017) 093210 and therefore not a novel entropy estimator. Thus, as the
authors have already acknowledged in their reply, the main result of
their work is neither the average or trajectory-dependent entropy
estimator per se, nor a thermodynamic interpretation for the effective
affinity a of a transition in terms of "effective thermodynamics".
Rather, it is the successful application of martingales and the obtained
law for the infimum current which comprise the novel insights of this
work.

Arguably, this focus is not sufficiently emphasized in the present
version of the manuscript (especially in the abstract and introduction).
It becomes clear only after carefully comparing this work to the relevant
references.

Despite this minor shortcoming, I recommend publication of this work
in SciPost as now is.

---

## Round 3 · Referee Report · Anonymous (Referee 2) · 2023-2-16

Report

I'm sorry for the delay, I had overlooked that this manuscript needed my input.

With the changes in the manuscript and the replies to the previous comments (almost) all questions have been clarified, and the manuscript should now be ready for publication.

Just a few optional points to consider, and responses to the answers to my previous points:

(d): I had kind of expected that proportionality is understood up to boundary terms. I'm not aware of any good examples of non-equilibrium systems where the entropy production is indeed proportional to a single edge current. This would mean that the entropy production of all other edges is zero exactly (otherwise the entropy would sometimes change without a change in Jxy). But if there is a current through one edge, there must also be a current though some other edges (by the Kirchhoff rules, see also point (f)), which comes with entropy production. The only example I can think of is a network with a single state and a single edge, leading from the state into itself. This somewhat stretches the present notation, where I believe x!=y is usually assumed out of notational convenience. If this asymmetric random walk is indeed the only relevant example, then I think Sec. 3 would profit from being specified for this system. On the other hand, the class of examples could be enlarged by adding dead-end edges that carry no current. At least I'd like to see discussed that the class of systems where this proportionality is exact is quite limited.

(f): I appreciate the clarifications concerning the "driving by a single edge". However, I feel that Sec. 6.4 now carries very little information, if any. Isn't this just introducing more notation, essentially defining the "thermodynamic force" simply as the microscopic affinity divided by temperature? Either the fundamental relevance of this class of systems should be explained in Sec. 6.4, or the Section could be removed altogether, in favour of simply pointing out the possibility of driving by a single edge in the context of the examples.

(h): I was oblivious of the fact that the quadratic bound [Eq. (16) of the reply] could yield stronger bounds on the entropy production than the TUR, when evaluated at non-typical z. Thank you for pointing this out. So am I right to assume that Eq. (16), evaluated at the optimal z, would, in Fig. 5, perform just as well as (or even better than) the estimator based on the infimum statistics? (I don't expect the authors to add this to the plot.) Nonetheless, I see the advantages of their new formalism.

  1. Thank you for this detailed and illuminating explanation. Though I feel I should have consulted a textbook rather than bothering the authors ...

---

## Round 3 · Author Response

Dear Editor,

Both Referees commented positively on the manuscript, for example, both Referees mention that the validity, significance and originality of the paper is high.

We have taken the suggestions of the Referees to improve the manuscript onboard, and we have replied to all their queries (see individual replies to the Referee's comments).

Having taken onboard all Referees' comments, we hope the present manuscript will be considered suitable for publication in SciPost Physics.

Kind regards,
Izaak Neri and Matteo Polettini

---

## Round 3 · List of Changes

To address the Referees' comments, we have made, amongst others, the following changes in the manuscript:

*) Following the First Referee's suggestion, we have added a paragraph in the introduction discussing the relation between the submitted manuscript and the paper [G. Bisker, M. Polettini, T. R. Gingrich and J. M. Horowitz, Hierarchical bounds on
entropy production inferred from partial information, Journal of Statistical Mechanics:
Theory and Experiment 2017(9), 093210 (2017)], which is Reference [35] in the submitted manuscript.

*) We have clarified in the abstract that estimates of dissipation based on infimum statistics are better than a naive estimator of the Kullback-Leibler divergence that neglects nonMarkovian correlations in the trajectories of the current.

*) We have added the Appendix C that compares microscopic affinities with effective in unicyclic systems. This is to address a comment of Referee 2 regarding the relation between the microscopic affinity, the effective affinity, and the "true" macroscopic affinity.

*) We have added Appendix H that comments on the infimum statistics of currents that are not edge currents (as requested by the second Referee).

---

## Editorial Decision

published